# A custom-made AAV1 variant (AAV1-T593K) enables efficient transduction of Japanese quail neurons in vitro and in vivo

Shaden Zoabi[1], Michael Andreyanov [1], Ronit Heinrich[1], Shaked Ron [1], Ido Carmi[1], Yoram Gutfreund[1] & Shai Berlin [1]✉

The widespread use of rodents in neuroscience has prompted the development of optimized viral variants for transduction of brain cells, in vivo. However, many of the viruses developed are less efficient in other model organisms, with birds being among the most resistant to transduction by current viral tools. Resultantly, the use of genetically-encoded tools and methods in avian species is markedly lower than in rodents; likely holding the field back. We sought to bridge this gap by developing custom viruses towards the transduction of brain cells of the Japanese quail. We first develop a protocol for culturing primary neurons and glia from quail embryos, followed by characterization of cultures via immunostaining, single cell mRNA sequencing, patch clamp electrophysiology and calcium imaging. We then leveraged the cultures for the rapid screening of various viruses, only to find that all yielded poor to no infection of cells in vitro. However, few infected neurons were obtained by AAV1 and AAV2. Scrutiny of the sequence of the AAV receptor found in quails led us to rationally design a custom-made AAV variant (AAV1-T593K; AAV1*) that exhibits improved transduction efficiencies in vitro and in vivo (14- and five-fold, respectively). Together, we present unique culturing method, transcriptomic profiles of quail's brain cells and a custom-tailored AAV1 for transduction of quail neurons in vitro and in vivo.

[1] Department of Neuroscience, Ruth and Bruce Rappaport Faculty of Medicine, Technion- Israel Institute of Technology, Haifa, Israel. ✉email: shai.berlin@technion.ac.il

Avian species are a powerful experimental organism in neuroscience (e.g., refs. [1–7]). Birds display sophisticated cognitive capabilities and specialized behaviors, such as long-distance navigation[8], imprinting, homing, food-caching, song learning, etc.[9,10]. Interestingly, these complex cognitive capabilities are presented by birds despite their diverging neuroarchitecture and, at times, suggested lower neuronal densities in comparison to mammals[4,11]. These thereby provide unique opportunities for comparative studies of cellular mechanisms leading to behavior[3]. Indeed, these capabilities and behaviors are behind the increased interest in avian neuroscience in recent years (e.g.,[1,2,7,9,12–20]).

Of the various avian species commonly employed today, we focused our attention on the domestic Japanese quail (*Coturnix japonica*). Quails are relatively small animals and therefore require smaller housing and animal facilities, reach sexual maturity quickly (significantly shorter than mice and chicken) and female quails lay approximately one egg per day; ideal for routine experimentations[15]. A particular benefit of this model over other birds is their ground-dwelling nature, which may simplify various behavioral studies (such as spatial navigation[20]) by reducing dimensionalities (e.g., ref. [21]). Despite the latter, (and despite their extensive use in developmental biology[15,22,23]), quails are rarely used in neuroscience[6,20]. In part, we deem this to result from a lack of tools and methods for monitoring and examining neurons in quails in vivo[21]. More precisely, whereas mammalian neuroscience heavily relies on viral tools for the delivery of genetic optical probes for interrogating the brain (e.g., GCaMP and Channelrhodopsin; ChR)[24,25], the use of viruses in quails has been primarily, if not exclusively, employed for transgenesis (and almost exclusively by use retroviruses, Lentivirus and MoMLV)[6,15,26–31]. Thus, whether transduction of other cell types at different developmental stages (including the adult animal) can be obtained by lentivirus or other viruses (e.g., adeno-associated virus; AAV) remains unknown[15]. Notably, the shortage of viral tools is not limited to the quail model; rather is a recurring theme in the avian field (refs. [32–35], but see below).

We sought to bridge this gap by screening for suitable viral vectors for the transduction of neurons of Japanese quails. To screen for numerous viral candidates, we develop a protocol for culturing primary neuronal cultures from quail embryos. Systematic characterization of cells in cultures demonstrates that the cultures are viable and contain over 11 different cellular populations, including two populations of mature neurons. We use the cultures for screening of multiple viruses commonly used in rodents and other avian species, although find that none yield efficient transduction, especially not as obtained in rodents. We then revert to rational engineering of viral AAV-capsids, which leads to the development of a single infectious AAV1 variant (AAV1-T593K; AAV1*) that exhibits significantly improved infection efficiency in vitro (eightfold). This variant also proved suitable for the transduction of primary chicken cultures. Lastly, and importantly, we found that AAV1* yields an approximately fivefold improvement of transduction of quail neurons in vivo.

## Results

**AAV1 poorly infects quail neurons in vivo.** It is well appreciated in the avian field that viral transduction of neurons in vivo is challenging[32]. However, a handful of recent reports show partial success following the use of common recombinant adeno-associated viruses, for instance, AAV1 in pigeons[32], barn owls[14] and zebra finches[36,37] (and other serotypes in canaries[7], finches[38], and more[39,40]). Thus, we first examined whether AAV1 would also be suitable for the transduction of neurons of Japanese quails in vivo. We produced and injected YFP-expressing AAV1 (see

viral titers and injection procedures in Methods) into the Wulst[41] of two months old quails, followed by cryosectioning and fluorescence imaging of brain slices (without immunostaining for YFP, see Methods) over the course of three to 8 weeks. Unfortunately, we found weak expression in one animal (and only at 7 weeks post viral injection) (Supplementary Fig. 1). The poor infection efficiency and extreme variability between the different animals could not be explained by factors such as different virus batches or virus viability, as each virus (whether made in-house or purchased) was validated on cultured cells (including primary rodent neurons). Of note, although it is plausible that detection of YFP-fluorescence could have been enhanced by immunostaining for YFP, we were not inclined to do so as our intentions were to obtain sufficient expression that would also be suitable for in vivo imaging. Thus, these results deterred us from continuing our explorations with added viral serotypes in the same manner.

**Development and characterization of primary cultures from quail embryos.** To rapidly screen multiple different viruses, we sought to examine in vitro transduction of cultured cells. To produce primary neuronal cultures, we explored common culturing protocols from rodents and chickens (we found no mentions for quails)[33,42,43]. Briefly, we dissected forebrains from seven to nine days in ovo (DIO) embryos or post-hatched chicks, mechanically dissociated and enzymatically digested the tissue, followed by plating of cells onto poly-D-lysine (PDL) covered glass coverslips; grown in standard growth media and incubator conditions (see Methods and Supplementary Table 1). Our rationale behind this staging was based on reports suggesting that neurogenesis should be completed by seven DIO[44,45], thereby yielding neuronal cultures with low amounts of mature glial cells[46]. During the first hours after plating, we could distinguish cell bodies, some showing extending neurite-like processes (Supplementary Fig. 2a, arrowheads), but only in cultures produced from embryos. Cultures from post-hatched chicks did not yield viable cultures. Nevertheless, embryonic cultures progressively waned, and most cells died past ~four days in vitro (DIV) (Supplementary Fig. 2b, c). We next examined growing the cultures directly on the tissue culture plate itself, as well as substituted the MEM-based growth medium (frequently employed in rodent and chicken culturing protocols[33,46]) to an enriched Neurobasal-medium (as employed for rodent organotypic slices[47]) (Supplementary Table 1). These collective modifications allowed the cultures to thrive, particularly cultures produced from 9 DIO embryos, and these remained viable past three weeks (Fig. 1a, b). Cultures produced from an entire hemisphere (plated onto 60 mm culture plates) yielded a rich network of highly interconnected and branched cells, most of which (>90%) showed positive NeuN[48]- and NeuroTrace-staining[49,50] (Fig. 1c and Supplementary Fig. 2d), although a much smaller fraction of cells displayed mature neuronal morphology (Fig. 1b, arrowheads). MAP2-staining indicated a lower estimation (~18%) of mature neurons in the culture (Fig. 1d) (discussed below).

To assess the culture's viability and to confirm the presence of neurons in the culture, we patched cells with neuronal morphology (e.g., Fig. 1d, inset) and, indeed, these exhibited robust action potential firing (under current-clamp) and, under voltage-clamp conditions, displayed prototypical barrages of excitatory postsynaptic currents (EPSCs) (Fig. 2a). We found that several electrophysiological features of quail neurons are highly similar to those of age-matched cultured neurons produced from mouse embryos (Fig. 2b and Supplementary Fig. 3a), albeit exhibit significantly lower intrinsic excitability (Fig. 2c, d and Supplementary Fig. 3b, c). Thus, our culturing protocol yields viable cultures with various cell types; however—

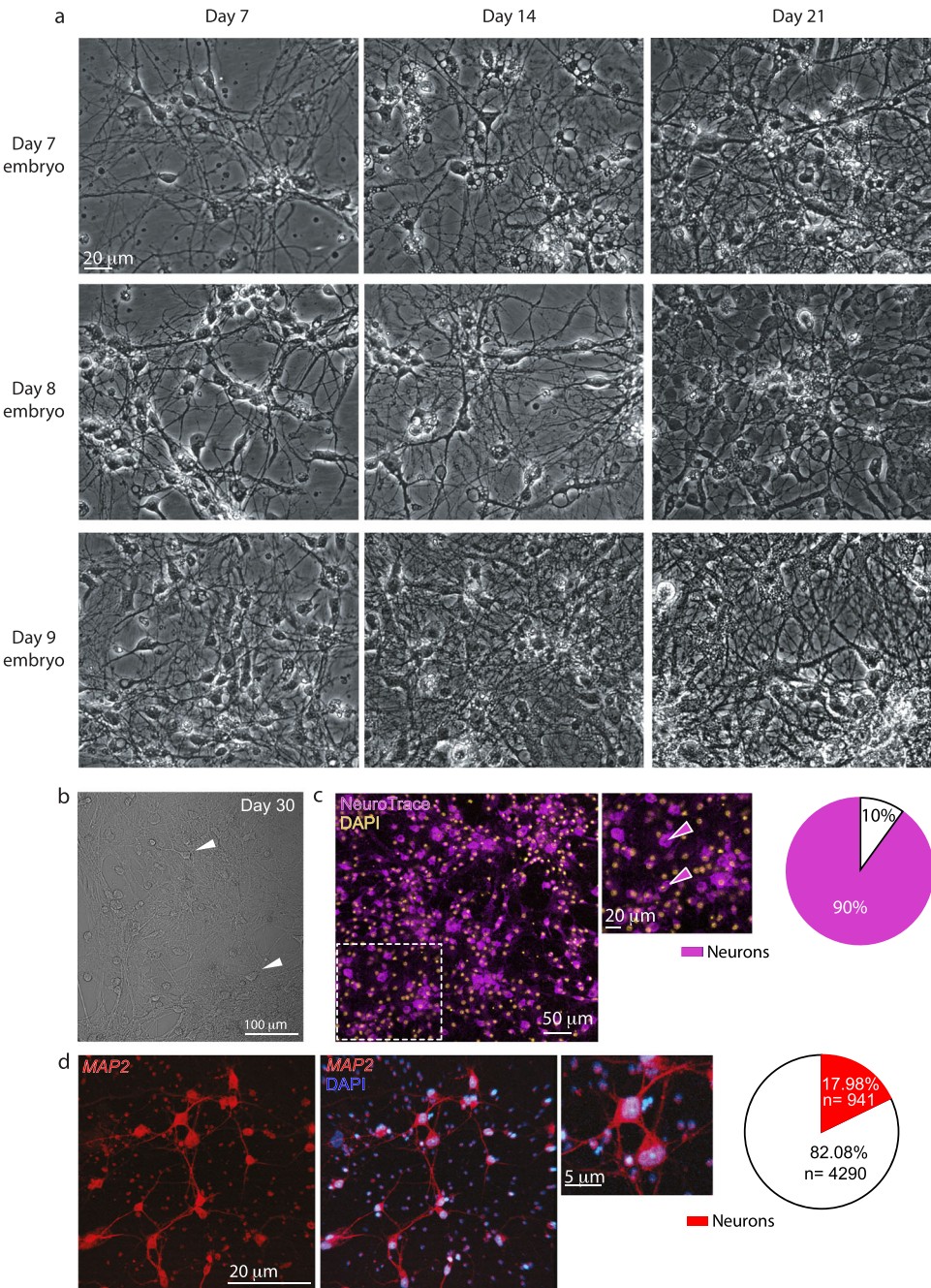

**Fig. 1 Development of primary cultures from embryonic quail brains. a** Micrographs of cultures produced from 7, 8, or 9-day-old embryos (days in OVO, DIO) (rows), grown for 7, 14, and 21 days in vitro (DIV) (column). **b** Cultures remained viable for up to 30 days (arrowheads represent examples of viable cells). **c** Seven DIV cultures (produced from 9 DIO embryos) co-stained with DAPI (yellow) and NeuroTrace (pink) suggest that the majority (~90%) of cells (inset: pink arrowheads) represent neurons, with a smaller fraction (10%) of non-neuronal cells, summarized in the right most panel. **d** MAP2-staining of cultures suggests that ~18% of cultured cells are neurons (summarized in the pie chart).

and importantly—the various staining techniques suggest that the cultures include ~20% mature neurons.

**Single-cell mRNA sequencing of embryonic quail cultures.** To examine the different cell types obtained in culture, we employed single-cell mRNA sequencing (sc.mRNA seq.[51]) of cultured cells produced from 9 DIO embryos (grown for 7 DIV; Methods[52–54]). Unsupervised clustering of the data (based on the significant top ten or top 200 differentially expressed-genes; DEGs) acquired from ~10,000 cultured cells revealed 15 noticeable clusters of cell types (clusters #0–14); all clusters exhibiting a similar (and high) number of unique molecular identifiers (UMI or nCount) and gene counts (nFeature) (Supplementary Fig. 4a–d and Fig. 3a, b). Importantly, all cells exhibited an acceptable mitochondrial genes ratio (10–20%)[55,56], demonstrating the viability of cells in cultures, and these were on par with sequencing results obtained from the brains of chicken embryos[45] (Supplementary Fig. 4a, b).

Before proceeding to cell annotations, many transcripts had to be manually curated (e.g., Fig. 3b, red annotations), although dozens of other genes could not be identified despite multiple reference genomes (e.g., ensembl.org), including the recently

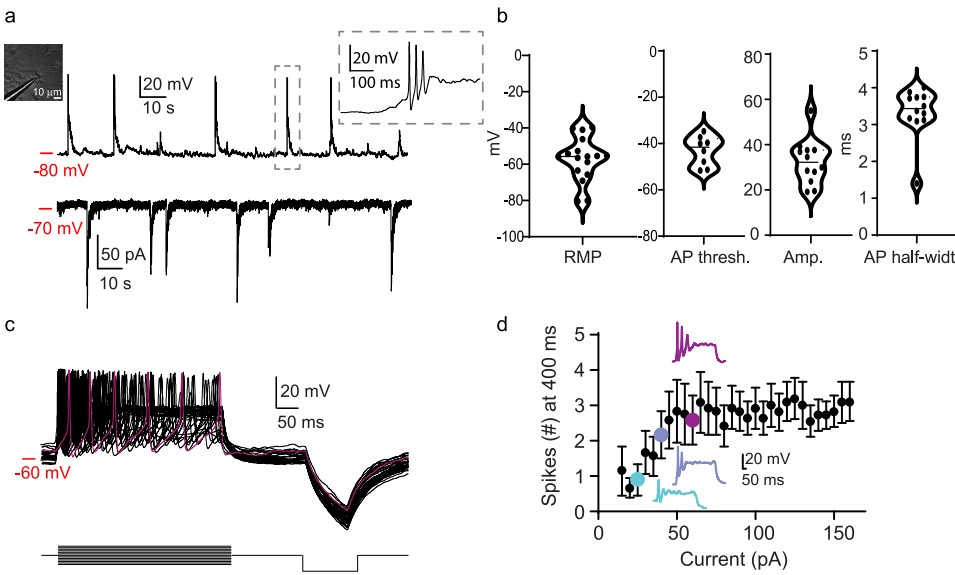

**Fig. 2 Electrophysiological description of primary cultured quail neurons. a** Representative current-clamp (top) and voltage-clamp (bottom) recordings from cultured neurons (left inset—micrograph of the patched cell). The current-clamp recording shows the resting membrane potential of the neuron (−80 mV), excitatory postsynaptic potential (EPSPs), and action potential firing (e.g., dashed rectangle and inset). During voltage-clamp, the cell was clamped at −70 mV, during which barrages of excitatory postsynaptic currents (EPSCs) could be distinguished. **b** Summary of electrical properties of quail neurons in vitro ($n = 12$). **c** Assessment of intrinsic neuronal excitability. Representative voltage traces (top) from a single neuron in response to current injection (bottom protocol) from which we could deduce the maximal firing rates, summarized in **d**. Data are presented as mean ± SEM. Color-coded traces correspond to colored data points on the plot (For the complete dataset, see Supplementary Fig. 3c).

published (though partially-sequenced) quail genome[57]. Identification and classification of cell type was done by querying multiple cell-type specific or enriched gene markers, as previously described[58–60] (along online atlases, e.g., proteinatlas.org). The largest group of cells (2400 cells) displayed a broad and dispersive expression of most top DEGs (Fig. 3a, b—light brown; subclusters are shown in Supplementary Fig. 4c–e). This cluster (Cluster #0, Supplementary Fig. 4d) showed a combination of neuronal (*STMN2, CDH2*) and glial markers (*FABP7, PROM1,* and *SLC1A3*), but also moderate expression of NES (neuroepithelial marker) and VIM (radial glia marker), representing very early Radial glial cells expressing early neuronal markers; likely a type of neural committed progenitor cells that precede neurons, such as pro-neuronal radial glia (Fig. 3c, see summary in Supplementary Fig. Table 2)[61–63]. This observation is supported by our NeuroTrace and NeuN-staining results suggesting the very high abundance of seemingly neuronal cells (see Fig. 1 and Supplementary Fig. 2)[64], and by the presence of two additional (and different) bona fide radial glia populations in the cultures that are distinguished from this cluster (Fig. 3c). Cluster #1 could be best classified as apical radial glia owing to *PROM1* expression[65] (along other glial markers, namely *FABP7* and *SLC1A3*), whereas cluster #2 shows the expression of *NES* and dispersive expression of added markers (reminiscent of cluster #0), suggesting a slightly earlier differentiation state of non-apical radial glia[65]. We could easily distinguish two mature neuronal populations, both expressing specific pan-neuronal markers *STMN2, SYT1* (Fig. 3c, clusters 6 and 7), as well as *MAP2* and *NeuN* (*RBFOX3*) (Supplementary Fig. 4f). The inhibitory neuronal population exclusively expressed *GAD1* and *-2, DLX1* and *-2*, and moderate levels of *SOX11*, whereas excitatory neurons showed strong expression of *GABBR2, GRIN* (*1* and *2B*), and *EPN2* and, uniquely, *SLC17A6*. Interestingly, *NEUROD6* (a glutamatergic neuron marker[66,67]) was enriched in this cluster, but was also found at lower levels in inhibitory neurons (Fig. 3c). This transcriptome signature is less common in mammals (e.g., refs. [59,68,69]). We did not detect other neuron types in this brain

region. We further found three distinct microglia populations (clusters #5, 8, and 13, Supplementary Fig. 4d, e); one cluster (cluster #13, $n = 130$ cells) showing added proliferation markers, explicitly *HMGB2, CENPF, TOP2A,* and *CCNB2*[70,71]. These imply that cluster #13 represents dividing microglia (Supplementary Fig. 4e). These are consistent with reports showing the complex heterogeneity in microglia populations, especially during early developmental stages compared with the adult[72,73]. The smallest cluster of cells (#14, $n = 125$ cells) is distinguished as oligodendrocyte precursor cells owing to the expression of *PTPRZ1, PMP2,* and *PDGFRA* (Fig. 3c, PDGFRA; manually annotated). Cluster #10 lies at the interface between clusters 0, 4, and 14, with neuronal progenitor cells (NPC) markers and additional division markers as seen in cluster 13, specifically *HMGB2* and *TOP2A*[70,71], therefore depicting a non-quiescent (i.e., dividing) neuronal progenitor stem cell population. We also find that the cultures contain ~14% fibroblasts (clusters # 3, 9, 11; $n = 1415$ cells) by the variable expression of *COL1A1*[74], arising from remaining meninges and perivascular system[75]. Interestingly, one fibroblast population (#11) showed a very weak expression of neuronal markers (e.g., *SOX11* and *ENC1*) (subcluster shown in Supplementary Fig. 4d, e). This is a unique scenario for which we could not find any mention in the literature. In fact, the inclusion of *SOX11* (a transcription factor) in fibroblasts (postnatal skin fibroblasts) is used to differentiate the cells into cholinergic neurons[76]. Identification of clusters #12 and 4 proved more challenging. Cluster #4 (residing adjacent to clusters 0–2) expresses RGs markers (*SLC1A3, VIM, NES, PROM1*; similar to cluster #1), along with expression of *MFGE8, GJA1,* and *ID3*; astrocytic markers in the mouse[60,77]. However, this cluster (or any other) presented no other astrocytic markers, notably *GFAP* (Supplementary Fig. 5a)[78]. This deficiency was further confirmed by the lack of immunostaining for the protein, whereas control rat cultures showed extensive labeling (Supplementary Fig. 5b, left and middle panels, c). Furthermore, cluster #4 shows weak, albeit significant, expression of *RELN* (neuronal marker). This unexpected transcriptome signature led us to

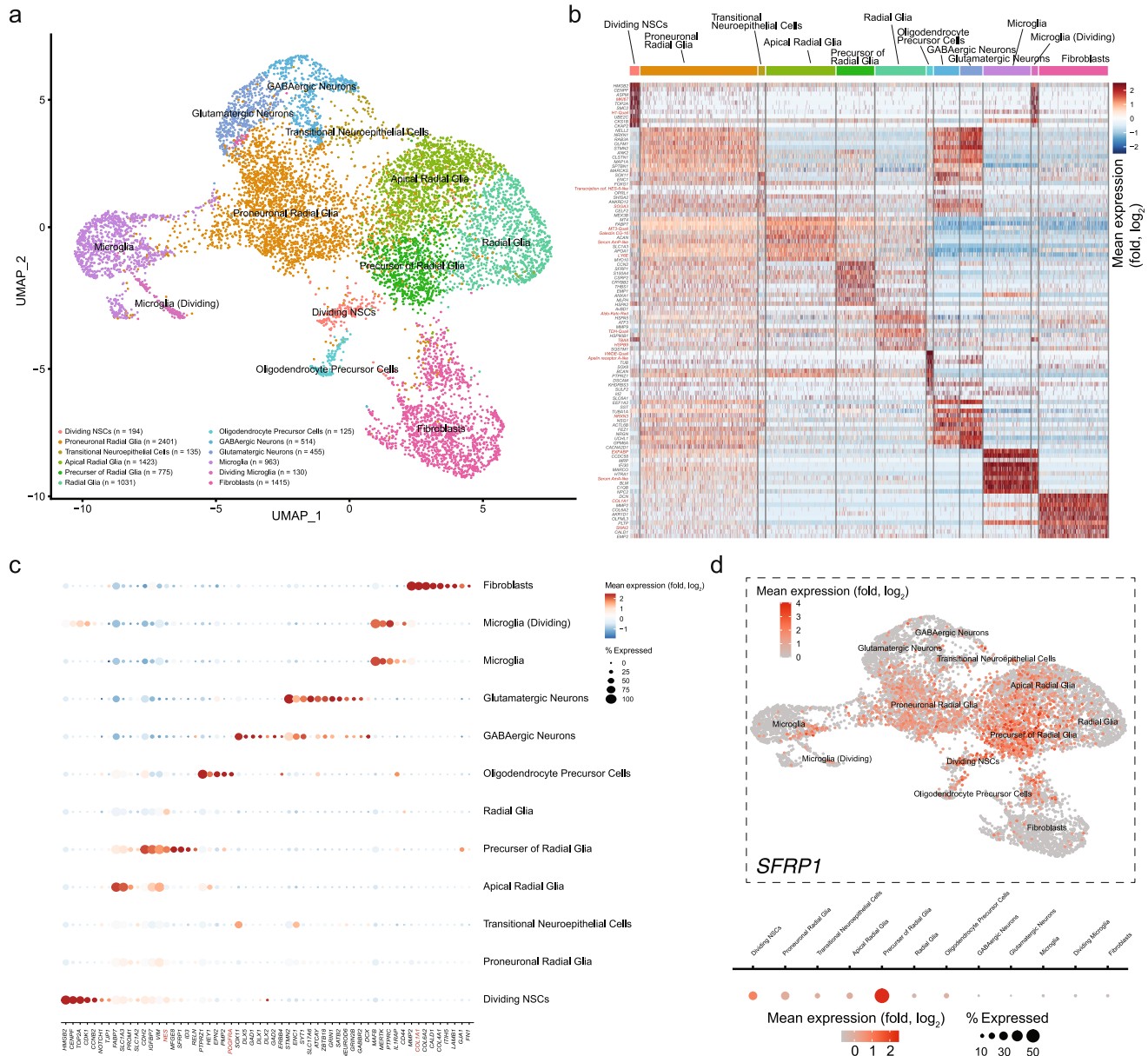

**Fig. 3 Single-cell mRNA sequencing of cultured cells reveals a large number (11) of cellular populations. a** UMAP plot of 9561 cells, taken from cultures produced from an embryo's entire hemisphere, classified based on the top ten most differentially expressed genes (DEGs), as shown in the heatmap in (**b**). The number of cells within each cluster are noted at the left corner of UMAP and clusters' colors correspond to column colors in **b**. Red annotated genes (in **b**) were manually curated (i.e., identity was not automatically found in reference genomes). **c** Cell type identification dotplot shows mean expression levels (blue to red) and percent of cells (size of data point) in each cluster and for select markers (bottom axis). Red annotated genes were manually curated. **d** Feature plot showing expression of SFRP1. Mean expression (Log_2 fold change) and percent of cells for each cluster are depicted below the plot.

hypothesize that this population represents an earlier differentiation state than RGs, namely RG-Progenitors (RGPs)[79,80]. Indeed, ~65% of cells within cluster #4 showed enriched levels of SFRP1; a prominent marker for RGP (Fig. 3d). Lastly, cluster #12 (n = 135 cells) shows strikingly overlapping transcriptional signatures with cluster #0, but with higher expression of inhibitory neuronal markers (*SOX11* and *ENC1*), though with no other GABAergic markers. Its location between clusters # 0, 1, and 6 suggests it to represent a slightly more NPC-RG differentiated state toward inhibitory neurons[81]. Together, we find that the quail embryonic cultures provide a very rich repertoire of cell types[82], ideal for studying stem cells of the brain. Notably, despite the embryonic origin of our cultures, we find very low amounts (and in very few cells scattered across

all clusters) of *SOX2* and *PAX6*—markers of very early developmental stages as reported for the chicken embryo (Supplementary Fig. 6)[45] (and see discussion).

**Viral screening in vitro and rationale evolution of viral capsid to infect quail brain cells.** We proceeded to screen for transduction of primary quail cultures by various viruses. We focused on common adeno-associated viruses currently in use: AAV1, AAV2 (and variants), AAV9, AAV-PhP-B[83], Baculovirus (baculo.)[84], lentivirus (lenti.)[26,27], and an avian AAV (A3V)[33]. We produced YFP-expressing viruses (AAVs, A3V and lenti.) and purchased a commercial baculovirus expressing GCaMP6 (genetically-encoded calcium indicator[85]) (see Methods). We infected the cultures in parallel to cultured mammalian cells

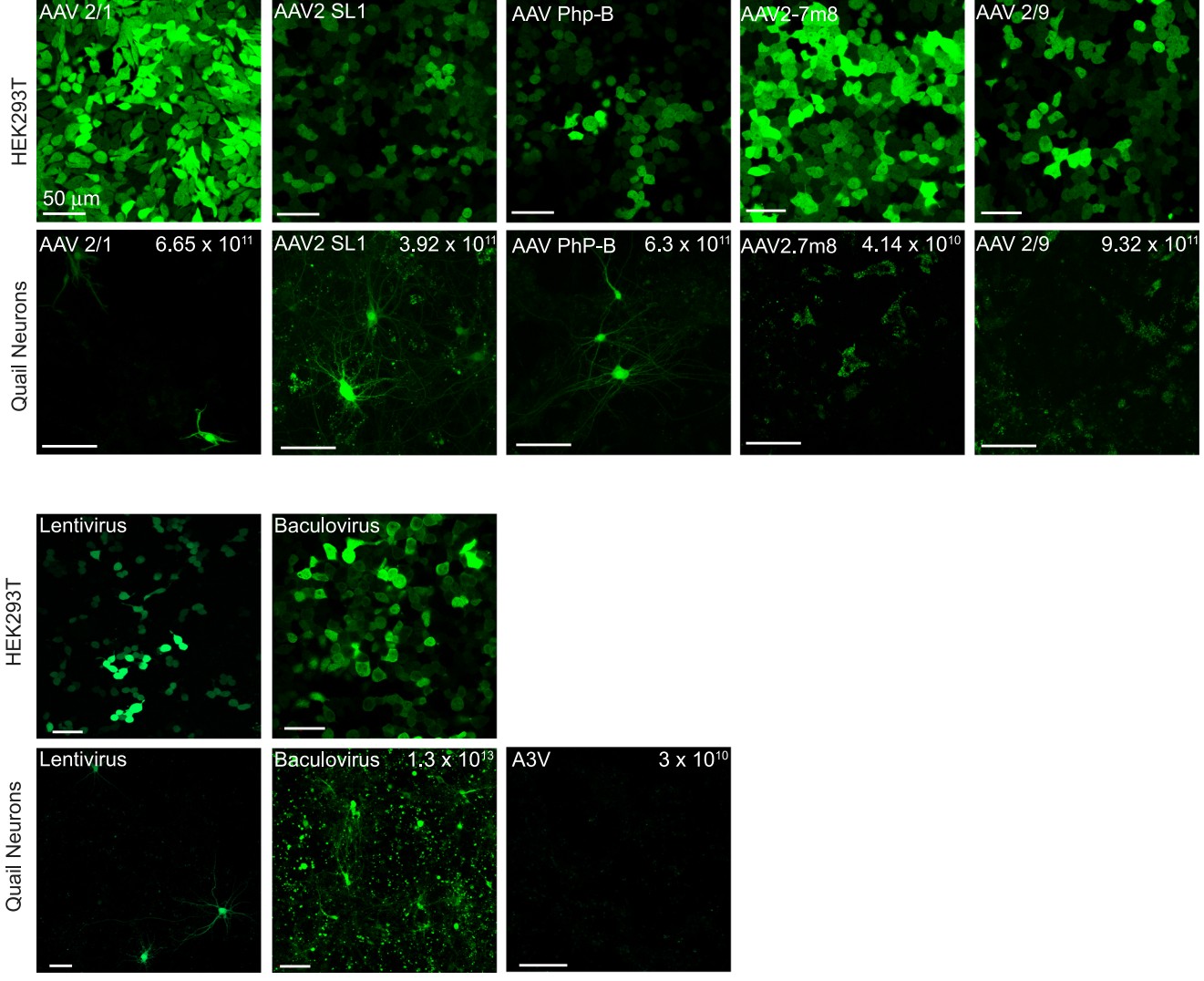

**Fig. 4 Common viral tools currently used in neuroscience poorly transduce primary quail cultures.** Various AAV serotypes, lentivirus, and baculovirus show very limited transduction efficacy of quail cultures. Infection efficacy was dually assessed on cultured cells (HEK293T cells, top lines) and quail cultures (bottom lines). AAVs (including A3V) express CAG-eYFP, Baculovirus expresses CAG-jGCaMP7f, and the lentivirus expresses CMV-eGFP. The strong expression of fluorescent markers within HEK293T cells (except for A3V, which showed no infection of HEK293T cells[83]) demonstrates the viability of the viruses used. Despite the latter, very few infected cells (mostly neurons) were detected in quail cultures. Note the cytotoxicity of the baculovirus.

(HEK293t cells), followed by an assessment of fluorescence after three DIV (Fig. 4a). Whereas all viruses robustly infected and yielded expression in mammalian cells (except A3V, which showed no infection of HEK293t, as described earlier[33]), we observed very sparse infection and weak expression in quail cells. Of note, the lack of infectivity of our cultures by A3V is surprising because of its reported infectivity of chicken brain cells[33]. We do not know the reason behind these observations; however, following trials with three different viral batches (DNAs were kindly provided by its developers, see Acknowledgements), we opted to stop pursuing this variant. Nevertheless, and importantly, a few positively-infected neurons (assessed by morphology) were observed following infection by AAVs, specifically AAV1, AAV2-SL1, and AAV-PhP-B and by lenti., though to a slightly lesser degree (Fig. 4). Baculovirus infection was highly toxic to cultured cells (Fig. 4, bottom row).

These low infection efficiencies motivated us to try to tailor these viral vectors for quail cells. To do so, we first examined the entry routes of AAVs into mammalian cells and noted that most AAVs do so by binding to non-specific proteoglycans at the membranes of cells[86]. However, AAV1 and AAV2 (and AAV5) also require the AAV-receptor (AAVR, also known as Dyslexia-associated *KIAA0319*-like protein) as a co-receptor for entry into cells[87–89]. Fortunately, our sc.mRNA seq. data revealed that our cultures express this transcript (at least to the method's detection limit), especially in microglia and neurons (Fig. 5a). The AAVR contains five polycystic kidney disease (PKD) repeat domains, from which PKD2 is recognized by AAV1 and AAV2 (Fig. 5b)[87,88]. Sequences of the PKD2 domain from various species show very high homology (Fig. 5c), though the very few differences between the sequences of quail and mammalian AAVR (qAAVR and mAAVR, respectively) occur specifically in residues that are essential for the interactions with AAV1 and AAV2, namely Q432 and K464 (mAAVR numbering) (Fig. 5b, inset, c and Supplementary Table 3). Of note, the divergent residues in qAAVR do not appear to be completely random, but rather mirror the residues found in the viral capsid. For instance, whereas mAAVR contains a lysine (K464) that interacts with a threonine (T593) found in capsids of AAV1 protomers (Fig. 5b, green and magenta, respectively), the qAAVR has a threonine

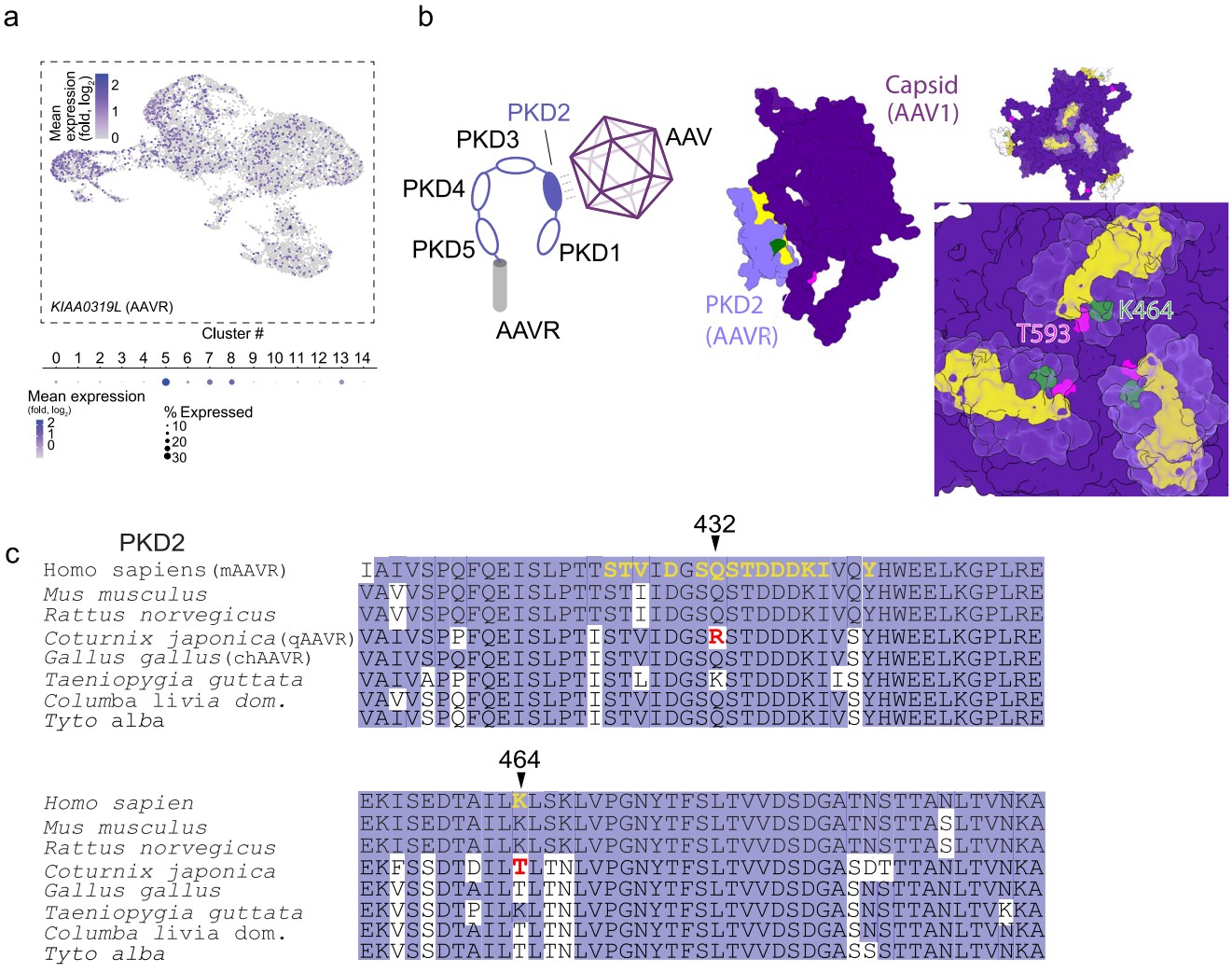

**Fig. 5 The AAVR is present in quail cells, but diverges in sequence from the mammalian receptor. a** Feature plot of KIAA0319L, the quail AAV-receptor (qAAVR). Mean expression (log$_2$ fold change) is noted at the left of the plot and the percent of cells per each cluster are depicted below the plot. **b** (left) Cartoon illustration of the binding of the AAVR (gray transmembrane domain and PKD domains in lavender) by an AAV (magenta icosahedron). The PKD2 domain is highlighted (dark lavender) because of its essential role in the binding the AAV's (AAV1 and AAV2) capsid[86,89]. (right) Atomic structure (by cryo-EM) of AAV1's capsid (magenta) bound to PKD2 (lavender) (pdb: 6JCQ[87]). Key residues in PKD2 that interact with the capsid monomer are highlighted in yellow. The K464 residue (green) in PKD2 interacts with residue T593 (pink) found in the capsid of a second AAV1 protomer. Inset: binding of the receptor (three yellow highlights) by multiple AAV capsid monomers (magenta) shows the very close proximity between K464 (green) and T593 (pink). **c** PKD2's protein sequence alignment between different species shows high conservation (lavender). Residues in the AAVR that directly interact with the AAV1 capsid are highlighted (bold yellow). Two essential residues differ between the quail and mammalian receptor (qAAVR and mAAVR, respectively)—R432 and K464 (bold red). Chicken AAVR (chAAVR).

(T204) instead. Similarly, while the mAAVR contains a glutamine (Q432) that interacts with an arginine of the AAV2 capsid, the qAAVR mirrors AAV2's capsid with an arginine of its own (R172). We thereby hypothesized that the mirroring residues (T-T for AAV1 and R-R for AAV2) may disturb the interaction between qAAVR and AAV1 and -2 viral capsids. To restore these interactions, we rationally mutated the residues in the capsids of AAV1 or AAV2 to match the residues found in the mammalian AAVR (instead of modifying the receptor, which would require transgenesis). Thus, this limited our mutagenesis towards residues that solely contain one interacting partner (Supplementary Table 3). We thereby produced AAV1 with a single mutation (T593K, denoted AAV1*) and two AAV2-variants (AAV2 and AAV2-SL1[90], an optimized AAV2 variant) bearing two mutations (R471Q and T592K), denoted AAV2* and AAV2-SL1*, respectively (see Supplementary Table 3 and Methods).

We initially infected primary rat neuronal cultures with AAV1* and found that infection efficacy was diminished approximately twofold (Supplementary Fig. 7a, b), supporting the importance of this residue for AAV's ability to infect mammalian cells[87]. We then infected the quail cultures with AAV1* and by AAV2* and AAV2-SL1* and found that, whereas the AAV2-variants yielded no infection of cells in multiple experiments (Supplementary Fig. 7c–e), AAV1* exhibited an order of magnitude (~14 fold) improvement in transfection efficiency of cultured cells (Fig. 6a). Moreover, AAV1*-infected cells showed approximately twofold brighter YFP-fluorescence (Fig. 6a, b). We patched YFP-positive cells with neuronal morphology (Fig. 6c, inset) and found that these exhibited very prominent action potential firing, robust synaptic activity, and expected resting membrane potential, as shown above (Fig. 6c, traces and see Fig. 2), whereas cells with non-neuronal

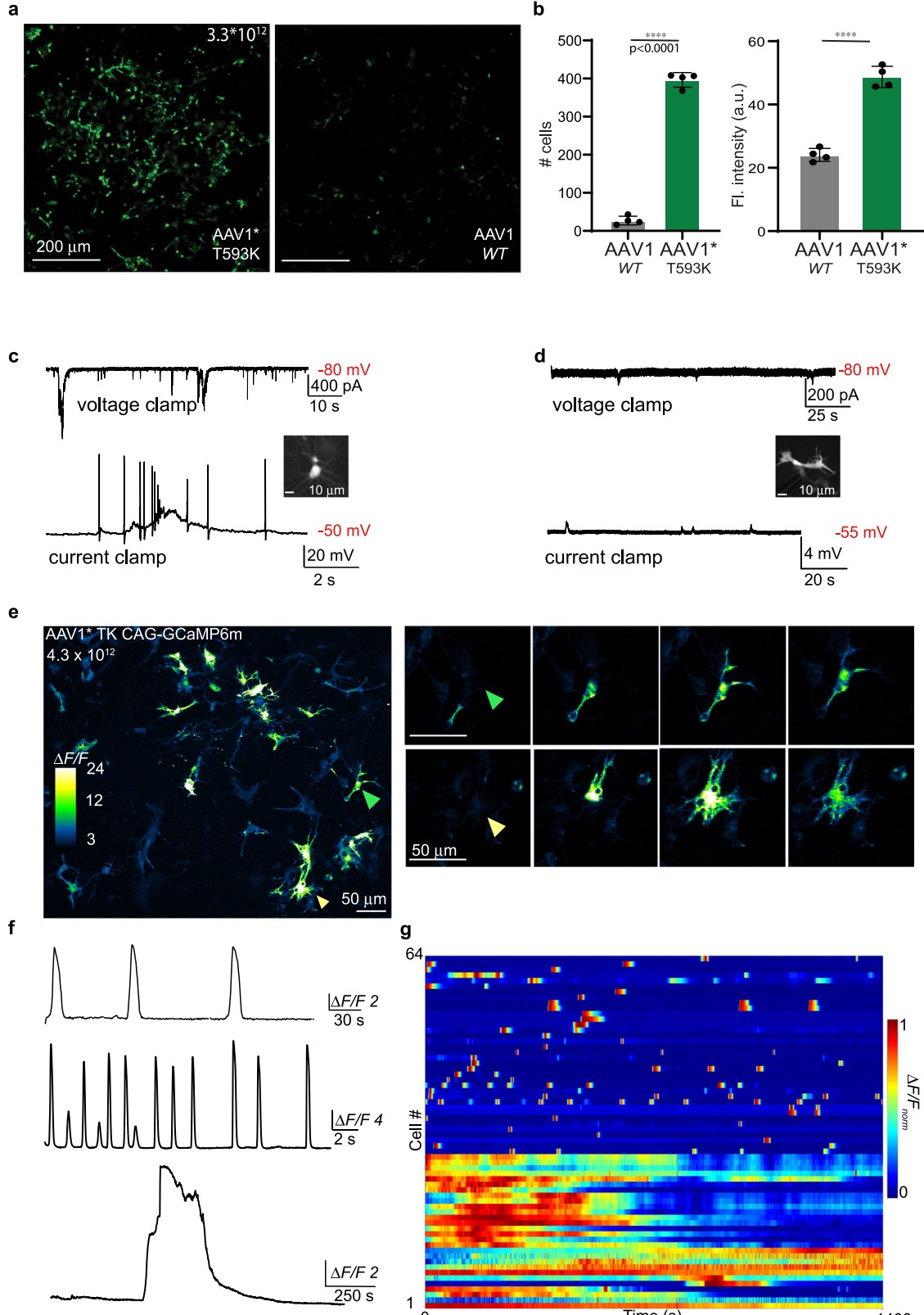

morphology (Fig. 6d, inset) did not fire action potentials, but maintained a healthy and hyperpolarized resting membrane potential, and displayed some electrical activities, reminiscent of glia cells (Fig. 6d, black traces) (e.g., refs. [91–93]). These demonstrate that AAV1* infected both neurons and non-neuronal cells (whether glia or neuronal precursors) and—

importantly—infection did not induce any detectable cytotoxic effects (attested by intact morphology and electrical properties), unlike infection by baculovirus (see Fig. 4).

We then packaged GCaMP6m[94] within AAV1* and infected 9 DIO cultures (grown for a week in vitro) to assess calcium activity in cultured cells. AAV1* transduction of GCaMP in these cells

**Fig. 6 The new AAV1 variant (AAV1\*) exhibits enhanced transduction efficiency of quail cultures and enables calcium imaging. a** Micrographs of quail cultures infected by AAV1\* (left) or by AAV1 (right) (both of identical titer—$3.3 \times 10^{12}$). The amount of YFP-positive cells (infected) and fluorescence intensity are summarized in (**b**). **b** AAV1\* infected significantly more cells compared to AAV1 by the same titer –AAV1\* (Mean ± SEM: 396.5 ± 9.5, AAV1 – 27.25 ± 5.7, T-test, $p < 0.0001$). Fluorescence intensity is significantly higher in cells infected with AAV1\* compared to AAV1 (AAV1\* - 48.7 ± 1.66, AAV1 – 24.06 ± 1.02, T-test, $p < 0.0001$). **c, d** Electrophysiological recordings of AAV1\*-infected cells (i.e., expressing YFP). **c-** Recording from a mature neuron (inset) reveals excitatory postsynaptic currents (top trace) and action potential firing and plateaus (bottom trace, $n = 5$), whereas amorphic cells (**d**, inset) do not fire (non-firing cells) and exhibit very health and hyperpolarized resting membrane potentials ($n = 4$). Small membrane activity could be easily seen, reminiscent of glial activities. **e** Micrograph showing cultures infected by AAV1\*-CAG-GCaMP6m and two representative cells showing propagating calcium activities (right). **f** Calcium-dynamics divided into three subgroups: oscillatory (top trace), spiking (middle), and long-lasting calcium plateaus (bottom), summarized in the heatmap in **g** ($N = 5$, $n = 64$). Data were presented as mean ± SEM, $p$ values are indicated, and noted by $*p < 0.05$; $**p < 0.01$; $***p < 0.001$; $****p < 0.0001$.

was not cytotoxic, and calcium activity could be easily monitored in many cells (Fig. 6e). We could distinguish a variety of different calcium-dynamics, such as slow and long-lasting calcium waves (reminiscent of glial activity[95,96]) and very sharp and transient $ca^{2+}$-spikes, likely action potential firing (Fig. 6f, g). Together, our results demonstrate the positive, though not exclusive[86], role of the pairing residues between AAVR and AAV1 for the infection of cells in vitro[87]. Importantly, AAV1\* shows improved infection efficiency of quail cells in vitro and is highly suitable for delivering optical probes, such as GCaMP6m, for monitoring activity from various cell types in culture.

**AAV1\* infects chicken neurons.** Quails are close relatives of chickens (*Gallus gallus*), which are a more widely used experimental model. We, therefore, sought to examine the infection efficiency of AAV1\* of chicken neurons. As noted above, chicken AAVR (chAAVR) only shows one divergent residue from mammalian AAVR (T464 instead of K464) (Fig. 5c). We first applied our culturing method to chicken embryos and found our method to be similarly suitable for this species, extending the usability of our protocol towards primary chicken cultures (Supplementary Fig. 8a). We infected chicken cultures with AAV1, AAV1\*, AAV2\*, and AAV2-SL1\*. AAV1 showed higher transfection efficiency of cells in chicken cultures, and as seen in quail cultures, AAV1\* significantly increased infection efficiency (Supplementary Fig. 8a–c, AAV1\*: 79 ± 16.7, AAV1: 42.7±4.4 cells/frame; $p = 0.055$, T-test). AAV2-variants were non-infectious (Supplementary Fig. 8d–f).

**In vivo AAV1\* transduction of neurons in young and adult quails.** To assess AAV1\* transduction in vivo, we injected AAV1\* encoding CAG-eYFP into the Wulst of five young (4–5 weeks old) quails and, in parallel, injected AAV1 encoding CAG-eYFP into the Wulst of five aged-matched quails. All injections were at similar depths, volumes, and titers (see Supplementary Table 5). After 7 to 8 weeks (waiting periods were based on previous reports[14,18,32,36,37] and see Fig. 1), brains were removed, fixed, and sectioned (Methods). Consistent with our in vitro results, AAV1\* showed significantly higher infection efficiency (~5-fold) compared to AAV1 (Fig. 7a–c; AAV\*— 509.2 ± 107.04, AAV1—101.9 ± 18.4 cells/mm², $p = 0.005$, T-test). This side-by-side comparison in young quails supports our initial observations following injections of AAV1 into adult quails' brains, in which instances we only found one animal to express YFP (Supplementary Fig. 1). Note that the highest density observed in animals injected with AAV1 was equivalent to the lowest density obtained by AAV1\* (Fig. 7c, compare with all examples provided in Supplementary Fig. 9). Infected cells also tended to exhibit higher YFP-fluorescence compared to AAV1, though this result did not reach significance (Fig. 7c, AAV1\*— 207.8 ± 5.9, AAV1—173.4 ± 12.04 a.u.). AAV1\* also showed better dispersion than AAV1 (Supplementary Fig. 8 and

Supplementary Table 4). Most infected cells appeared to be neurons (assessed by morphology and features such as spines), despite the ubiquitous promoter employed (CAG[97]) and the pan-tropism observed for AAV1\* in culture (see Fig. 6). Very few cells appeared glial (Fig. 7 and Supplementary Figs. 9, 10, arrowheads). Lastly, we also observed that injections of AAV1\* yielded expression in three adults (2 months old) quails, whereas AAV1 showed expression in only one animal (Supplementary Table 4 and Supplementary Fig. 9). Together, these observations demonstrate that AAV1\* significantly outperforms AAV1 for delivery of genetically-encoded tools to neurons of the Japanese quails in vivo, thereby making it—to the best of our knowledge— the first tailored-AAV for this bird species.

**Discussion**
Avian species provide unique opportunities for studying the brain (e.g., refs. [2,7,13,19]). The Japanese quail, in particular, presents several advantages over other avian species currently in use, for instance, faster sexual maturity, ground dwelling, and a high reproductive rate[15]. We have recently used this model to explore spatial navigation by use of electrodes, only to find that quails contain head-direction cells, as in other mammalian species, but we could not detect place or border cells in their hippocampal formation[20]. This is striking, as spatially-modulated cells are at the core of mammalian navigation and have also been recently described in another species of a food-caching bird[98]. We thereby suggest that this model provides distinctive opportunities to elucidate how the brain achieves spatial specialization across different species[99]. Despite the latter, genetic and molecular tools[15] (or transgenic lines[6,23,26–28]) for interrogating the brains of quails are much less abundant or completely absent[18,31,100]. These are behind the motivation to concentrate our efforts on the design of viral vectors for the transduction of quail brain cells.

Scrutiny of the literature reveals that the use of viruses for brain research in avian models is relatively uncommon[7,14,32–35,37], especially in comparison to mammalian models (e.g., refs. [83,101]). In the case of quails, there are even fewer descriptions, and in most, if not all, instances, retroviruses were employed for transgenesis[28,31,102–104]. Thus whether—and which—viruses can directly infect and transduce brain cells of the quail have yet to be demonstrated. We were particularly interested in testing whether AAVs can be used for the transduction of neurons in quails, for the reasons that AAVs are small, replication-deficient DNA viruses (safe to handle) and, despite their limited payload (<4.7 kb), are minimally toxic to cells, provide long-lasting expression of the gene(s) and can be produced at very high titers (~$10^{13}$)[101]. Importantly, many AAV serotypes show better spread in the brain and improved tropism towards neurons compared to other viruses (e.g., lenti.). To quickly sift through a long list of viruses, we developed a primary culturing protocol from the brains of quail embryos (Fig. 2). We chose the E9 developmental stage (equivalent to E10 or HH36 in chicken[105]) for producing the cultures based on previous reports that have

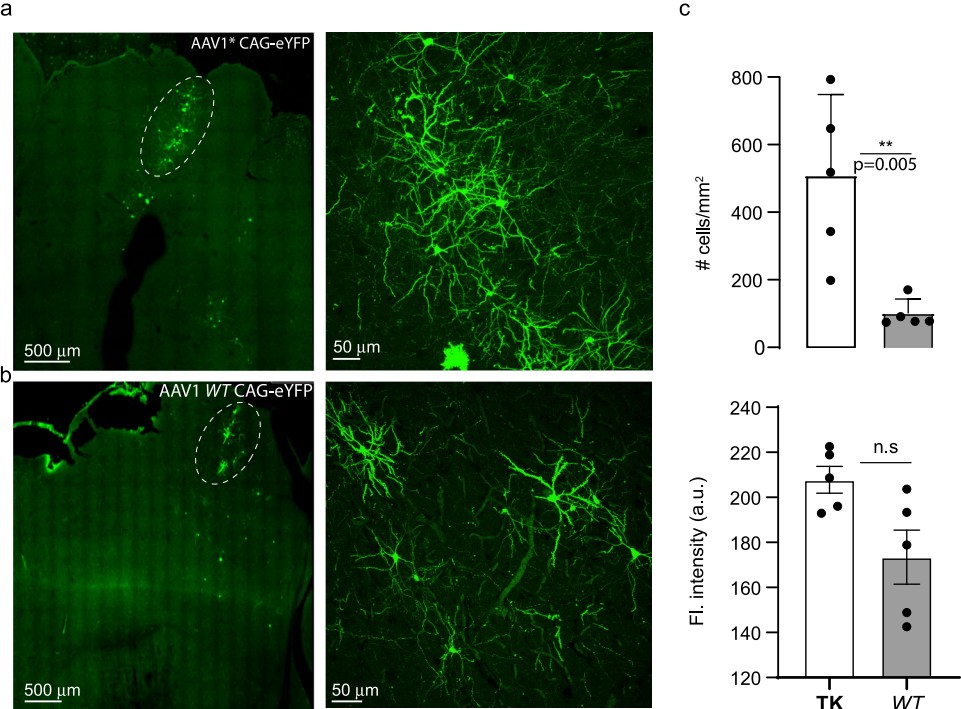

**Fig. 7 AAV1\* outperforms AAV1 in vivo in young quails. a** Coronal brain slices of young quails (4 weeks old), produced after 7 weeks from animals injected, side-by-side, with 1 µl of AAV1\*.or AAV1 (**b**); expressing CAG-eYFP. **c** AAV1\* provides a higher density of infected cells (mostly neurons, top)- AAV1\* (Mean ± SEM: 509.2 ± 107.04, AAV1—101.9 ± 18.4, T-test, $p < 0.001$). AAV1\*-infected neurons also show a tendency for higher expression within each cells (bottom)- AAV1\* - 207.8 ± 5.9, AAV1—173.4 ± 12.04. Each data point represents one animal ($n = 5$). The complete data set is provided in Supplementary Fig. 9. Note that the highest density observed in animals injected with AAV1 is equivalent to the lowest density obtained by AAV1\* (**c**, top). Data were presented as mean ± SEM, $p$ values are indicated, and noted by \*$p < 0.05$; \*\*$p < 0.01$; \*\*\*$p < 0.001$, n.s. non-significant.

suggested that neurogenesis should be completed by this developmental stage[44,45]; in which case there should be many neurons with smaller amounts of mature glial cells[46]. We deemed this ideal, as we were mainly interested in examining the suitability of various AAVs to infect neurons. Nevertheless, and although our results do show very few mature glia cells (e.g., mature astrocytes and oligodendrocytes are completely absent; Fig. 3c and Supplementary Fig. 5a), we found a very large population of neuronal precursor cells instead, with a small proportion of mature neurons (GABAergic and Glutamatergic- 969/9561 cells; ~10%) (Fig. 3). These numbers are not reflected by the use of Neuro-Trace (a Nissl stain–commonly employed for staining neurons[49,50]), but are supported by the use of *MAP2*-staining (~18%) (Fig. 1c, d and Supplementary Fig. 4f). Thus, and although beyond the scope of this work, we suggest that Neuro-Trace may also stain neuronal precursor cells (our survey of the literature suggests that this has not been systematically explored). Together, our results indicate that neurogenesis is not complete by E9 in the quail and this is supported by reports showing the abundance of neuronal progenitors at similar embryonic stages in cultures from chicks[106], as well as by in vivo reports of chicken embryos in which neurogenesis peaks at ~E7, but continues past this stage as late as E12, in various brain regions[107]. Furthermore, Radial glia (i.e., progenitors of neurons[108]) can be observed in later developmental stages, as late as E11. However, most previous studies employed various staining techniques, unlike our sequencing results, and these are hard to compare side-by-side (especially since some staining techniques may be promiscuous-e.g., NeuroTrace; above and Fig. 1c). The only available sc.mRNA seq. dataset is of the closely-related chicken species, but of much earlier embryonic stages (HH7 or E1)[45]. Thereby, it is less suitable for comparison with the developmental stage examined in this

report. For instance, although we find some very early progenitor cell markers, such as *PAX6* and *SOX11*, in our datasets (Supplementary Fig. 6), these are found in very low amounts in our cultured cells compared to chicken embryos[45].

Lastly, a comparison of our sc.mRNA seq. data with reported transcriptomes from other animal models (e.g., refs. [58,61,65,69,109]) reveals differences between species, such as lack of astrocytes in our cultures, the abundance of precursors of RG[77], as well as a unique fibroblast population with the expression of neuronal markers (e.g., *SOX11*) (Supplementary Fig. 4d, e). Thus, embryonic quail brains appear to present a unique treasure trove of cell types, which may be of interest to developmental and comparative biologists.

**AAV1-infection of cells**. Quail cells in culture are somewhat resistant to infectivity by most viruses tested (Fig. 4). This observation was somewhat unexpected as (1) several AAVs have been shown to efficiently infect avian neurons in vivo[14,32], (2) A3V has been tailored for infection of chicken neurons[57], and (3) the specific lenti. we have examined has been previously employed for transgenesis of quail cells (albeit germ cells)[26]. The reason(s) behind this resistance is unknown but, in the case of AAV1 and AAV2, we suspected divergence in the receptor of the viruses, i.e., the AAVR[89].

It is well established that the infection route of AAV is highly complex and requires a variety of membrane proteins that serve as co-receptors for the virus[86,110,111]. However, some variants (notably AAV1, 2 and 5) also require the AAVR[89]. We mined our sc.mRNA seq. data and indeed find our cultures to express qAAVR in a variety of cell types, but at low levels (Fig. 5a). These observations are highly consistent with the receptor's reported

"Low cell type specificity" expression patterns (see full details in ref. [112]). Interestingly, transcripts of qAAVR are found at slightly higher levels in microglia (Fig. 5a). Nevertheless, even in this cluster, mRNA of this transcript is only detected in ~30% of the cells (Fig. 5a). Whether these low levels are representative of low protein levels is unknown, and we could not address this by immunostaining as this receptor lacks a suitable antibody. In fact, the only available antibody we could find (HPA072692) shows the expression of the receptor intracellularly and is unable to detect receptors on cell membranes[112]. These prevented us from addressing whether there is a correlation between receptor levels and AAV1*-infectivity.

Scrutiny of the receptor's protein sequence demonstrates that the mammalian and the quail AAVRs diverge at key residues responsible for the binding of the AAV1 and AAV2's capsids (Fig. 5c)[87–89,101]. We also note that different bird species have different AAVRs, and that these slight differences could be part of the reason why some AAVs may be suitable for one species, but not others[14,32,39,40]. In support, the AAV-receptor in Zebra finches is more similar to that of rodents (Fig. 5c), which coincides with reports showing the efficiency of AAVs in transducing Zebra finches' neurons[36–38].

Our mirroring mutagenesis scheme significantly improved the infection efficiency of AAV1 in vitro and in vivo, however, it did not enable AAV2-variants to infect the cultures (Figs. 6, 7 and Supplementary Figs. 6, 8). In vitro, AAV1* infected a variety of cell types (Supplementary Fig. 11a, b), and this is consistent with the notion that AAV1 is not exclusively neurotropic[113–118] (unlike other AAVs, for instance, AAV2[118]). Correspondingly, in vivo, AAV1* infects both neurons and glia, although we did observe a slight preference towards neurons (Fig. 7 and Supplementary Figs. 9, 10). Based on these observations, we expected AAV1* to better infect chicken cultures owing to the higher homology between the chAAVR and mAAVR and that the single divergent residue, precisely residue K464 (mammalian numbering, Fig. 5c) which is involved in interaction with the viral capsid (Supplementary Table 3). Indeed, AAV1* outperformed AAV1 in transducing cultured chicken cells (Supplementary Fig. 8c). Together, these strongly support the important role of the AAVR in the infectivity of quail brain cells despite the necessity for added co-receptors[89]. Together, these imply that our unique strategy (in contrast to viral evolution methods that require highly specialized labs[83]) should be compatible with tailoring AAVs for other bird species.

In conclusion, our study provides a detailed description of the development of a unique quail-tailored AAV1—starting from the development of a culturing protocol, through molecular characterization of the embryonic cellular landscape of the quail's brain and, finally, to rationale engineering of AAV1's capsid exhibiting improved transduction capabilities of quail's brain cells in vitro and in vivo. Our efforts thereby expand the available toolbox for interrogating the brains of a new animal model, which should likely increase interest in this unique avian model.

## Methods

**Animal ethics**. Animal experimentations were approved by the Technion Institutional Animal Care and Use Committee (permit no. IL-157-11-17 and IL-19-10-143) and all experiments strictly followed the approved guidelines.

**Primary quail neuronal cultures**. Neuronal cultures were prepared from forebrains of seven, 8- or 9-day quail embryos. In a laminar hood, shells from post-fertilized eggs were gently removed (by breaking the shell at its upper tip). Embryos were isolated from the yolk and placed in Dulbecco's Modified Eagle Media/Nutrient Mixture F-12 (denoted dissociation medium). Tissues were then enzymatically and mechanically dissociated as previously described in ref. [47,119], with several key modifications (see Supplementary Table 1). Briefly, skin, skull, and meninges were removed by scissors and tweezers, and forebrains were isolated and

placed in a 15 ml conical tube containing dissociation media enriched with Papain (30 U/ml) and DNAse 1 (57 U/ml) and placed at 37 °C for 30 min. Following incubation, the solution was gently removed without disturbing the forebrains. Digested forebrains were washed three times with PBS and after the last wash, PBS was completely removed and replaced by 2 ml of plating medium consisting of neurobasal medium (Gibco, Cat. 21103049) supplemented with 2% B-27 (Gibco, Cat. 17504044), 1% Pen/Strep and 0.25% Glutamax (Gibco, Cat. 35050061). Then, tissue was manually dissociated by gentle trituration of solution with a fire-polished glass pipette (x15), followed by a single trituration with a 1000 μl plastic tip (to ensure complete dissociation). This solution was then applied onto a 40 μm cell strainer (placed on a 50 ml conical tube and pre-washed by 1 ml plating medium). The additional plating medium was added to the strainer after passing the entire solution to elute the remaining cells from the strainer. The filtered solution, containing the dissociated cells, was transferred to sterile tissue culture grade plastic plates (Corning, 60 mm, Cat. 430166) precoated with PDL, and these were placed for one hour in an incubator (37 °C, 5% $CO_2$). Following incubation, the medium was removed and replaced by a fresh plating medium (prewarmed in the incubator). Plates were placed back in the incubator and half the media was replaced every 2 days by a fresh and prewarmed plating medium.

**Mammalian cell culture and transfection**. HEK293T (Human Embryonic Kidney cells, ATCC #CRL-1573) were maintained in DMEM (containing 10% FBS and 1% L-glutamine) in 100 mm Corning cell culture dishes. Cells were purchased from the American Tissue Culture Collection (ATCC) and are regularly tested for mycoplasma. These cells were used to examine the infectivity of viruses (see Fig. 4) and produce viruses (see below—viral production section).

**DNA constructs and mutagenesis**. Helper plasmid pAdDeltaF6 (expressing adenovirus E4, E2A, and VA; Cat. 112867); Rep/Cap plasmids—pAAV2/1, pAAV2/9, pAAV2/SL1, PHP-eB, 7m8 (Cat. 112862, 112865, 81070, 103005, and 64839, respectively), and Transfer plasmid—pAAV-CAG-eYFP (Cat. 104055) were purchased from Addgene. For the production of Avian adeno-associated virus (A3V), all plasmids (Helper, A3V rep/cap, Transfer-RSV-eGFP) were generously provided by Prof. Watanabe (Kyoto University Japan)[33]. For lentivirus production, we used pFUGW, the HIV-1 packaging vector Δ8.9 (pΔNR/8[120]), the VSVG envelope glycoprotein vector (pVSVG) and FR(GCaMP6S-p2A-nls-tdTom) as described earlier[26], and these were a kind gift from Prof. Lois C. (Caltech, USA). Baculovirus (BacMam) containing CAG-GCaMP7s was purchased from Montana Molecular. Ltd (Montana, USA). Point mutations in AAV rep/cap plasmids were introduced by PCR; using PFU polymerase (Promega, United States). The PCR program consisted of 18 cycles of 55 °C annealing temperature and 68 °C (for 20 min) extension. (For a list of primers—see Supplementary Table 6).

**Electrophysiology**. Patch clamp recordings were obtained by MultiClamp 700B and Digidata 1440 A (Molecular Devices), and performed as previously described in ref. [121]. Briefly, cells were voltage-clamped at −70 mV. Borosilicate glass capillaries (i.e., pipettes) were pulled to resistances of 3–7 MΩ and were filled with an internal solution containing (in mM): 135 K-gluconate, 10 NaCl, 10 HEPES, 2 $MgCl_2$, 2 $Mg^{2+}$-ATP, 1 EGTA, pH = 7.3. Recordings were done in extracellular recording solution containing (in mM): 138 NaCl, 1.5 KCl, 2.5 CaCl2, 10 D-glucose, 5 HEPES, 0.05 glycine, pH = 7.4. Gap-free recording protocol was used to assess spontaneous activity in the culture. For assessing intrinsic excitability, cells were current clamped by injected currents to −60 mV. The current steps were at 50 pA increments and the number of action potentials was calculated. We analyzed only complete action potentials (not differential potentials). Data were analyzed using the Clampfit software (Molecular Devices, USA).

**Histochemistry**. Quail cultures (grown directly on plastic plates) were fixed by 4% paraformaldehyde (PFA) for 15 min, permeabilized with PBS containing 5% FBS (fetal bovine serum) and 0.1% triton X-100. NeuN or MAP2-immunostaining were performed by overnight incubation of fixed cells with anti-NeuN antibody (mouse, clone A60, Millipore CAT# MAB377) (1:1000 in PBS) or with anti-MAP2 (mouse, cat # MA5-12826, Thermo Fisher) (1:500 in PBS) at 4 °C, with 3% FBS and 0.1% triton X-100. The next day, plates were washed three times with PBS and stained by secondary antibody (Rhodamine Red-X goat Anti-mouse IgG, Jackson Laboratories, CAT# 115-295-003) (1:200), in 3% FBS and 0.1% triton X-100 for 1 h at room temperature. Plates were then washed by PBS and stained with DAPI (1:1000). NeuroTrace 640/660 staining (Thermo Fisher) was performed as previously described in ref. [122]. Briefly, cultures were fixed as described above, permeabilized with PBS containing 0.1% triton X-100 for 10 min, washed, and followed by incubation for 30 min with PBS containing NeuroTrace stain (1:100). Then, cells were washed by PBS and stained with DAPI (1:1000). GFAP-staining (1:500) was performed by using anti-GFAP (mouse, clone G-A-5, Calbiochem, CAT# IF03L), followed by staining with a secondary antibody (1:500) (Alexa Fluor-488 goat anti-mouse IgG, Jackson Laboratories, CAT# 115-545-003).

**Virus production**. We have produced viruses by the iodixanol method, as we previously described in ref. [123]. Briefly, HEK293T (Human Embryonic Kidney cells, ATCC #CRL-1573) were grown in Dulbecco's Modified Eagle Media/Nutrient

Mixture F-12 (supplemented with 10% Fetal bovine serum and 1% L-glutamate) on 10 ml tissue culture plates (Corning, Cat. 430167), at 37 °C and 5% $CO_2$. Cells were grown to 50 to 70% confluency and transfected with three viral plasmids (Helper, rep/cap, and transfer) using Polyethyleneimine (PEI) at a ratio of 8.1, 5.4, and 13.5 ug of DNA, respectively. After 6–8 h, growth media was replaced by serum-free media with 1% Glutamax. Media was collected after 48 and 72 h following transfection (collected media was preserved in −80 °C). The collected media was then filtered and concentrated via the Iodixanol step gradient method[124] to a final volume of 100–500 µl. Viral titer was determined by qPCR. Titers below $10^{10}$ were discarded. Viral titers of the viruses used: AAV1 WT—$3.13 \times 10^{12}$, AAV1*TK—$5.25 \times 10^{12}$, AAV2-SL1—$3.92 \times 10^{11}$, AAVphpB—$6.3 \times 10^{11}$, AAV2-7m8—$4.1 \times 10^{10}$, AAV9—$9.3 \times 10^{12}$, Baculovirus—$1.3 \times 10^{13}$, A3V—$3 \times 10^{10}$.

**Cryosectioning**. After 4 to 8 weeks following viral injections, anesthetized animals underwent whole animal perfusion fixation, as previously described in ref. [125]. Briefly, 4% PFA was perfused to the entire body of the animal via the vascular system through the heart of the quail. Whole brains were then dissected and placed in 4% PFA overnight. The following day preserved tissue was treated with 30% sucrose solution and embedded in OCT for cryosectioning. Brains were sliced (40 µm sections) via a cryostat.

**Fluorescence imaging**. Imaging was performed on a Zeiss Laser Scanning Confocal Microscope (LSM-880 or 900-Airy2; Zeiss, Germany), as previously described in refs. [126,127]. Experiments were conducted using a water immersion objective lens 20x [a water Plan-Apochromat objective lens; 20x/1.0 DIC D = 0.17 (UV) VIS-IR M27 75 mm] with a focal spot diameter of 0.5 µm ($D = 1.2 \times \lambda/NA$). Brain slices were imaged serially and automatically tiled for reconstructing brain hemispheres. Live cells (neurons and glia) were imaged in a standard extracellular imaging solution containing (in mM): 138 NaCl, 1.5 KCl, 1.2 $MgCl_2$, 2.5 $CaCl_2$, 10 D-glucose, 10 HEPES, pH 7.4. GFP/GCaMP were excited using a 488 nm laser; Neu-561 nm; NeuroTrace- 640 nm, and DAPI- 405 nm. Cultures were imaged after 3 days from viral infection. Change in fluorescence ($\Delta F/F$) was calculated by (Ft−F0)/F0, where Ft is measured fluorescence (in arbitrary units, a.u.) at a given time t and F0 is initial baseline fluorescence, typically calculated from averaging the 10 first images, representing the basal fluorescence. In cells showing activity, baseline fluorescence was taken during times of inactivity (or at the trough of fluorescence signals). $\Delta F/F = 1$ describes an increase of 100%, equivalent to a twofold increase in fluorescence. We did not need to encounter near division-by-zero artifacts and bleaching was minimal (most times not observed). Therefore, no corrections were made to fluorescent signals. For analysis of the percent of cells transfected by various AAVs (e.g., Fig. 6b), we imaged multiple (>5) large fields of view, in which we calculated the percent of infected cells by dividing the number of YFP-positive cells by the total number of cells. We averaged the percentages from multiple experiments, namely from different and independent cultures. Classification of cell types was obtained by staining (DAPI and NeuroTrace, see above Histochemistry). NeuroTrace-positive were counted as neurons, whereas NeuroTrace-negative denoted non-neuronal cells. The total number of cells was calculated by counting DAPI-positive nuclei (bright blue staining) in each field of view. This was obtained automatically by the ImageJ software. Assessment of cell density (per $mm^2$) in vivo was determined as previously described in ref. [32], namely by imaging identical fields of view (i.e., identical sizes of images see Fig. 7 and Supplementary Fig. 8, typically 0.5 mm × 0.5 mm) in which we counted the number of visible fluorescent somata (we did not count processes as these could project to several planes of view). This was repeated in multiple animals, as noted in the legends.

**Stereotaxic viral injections in vivo**. Adult quails were anesthetized by a mixture of 2% isoflurane and 1% nitrous oxide. A small craniotomy was made above the Wulst (on modified stereotaxic equipment), followed by the removal of dura by tweezers and injections of 0.5 or 1 µL of each virus (at 50 nl/10 s) at multiple depths. Craniotomies were sealed by silicon and C&B MetaBond (Parkell, USA), as previously described in ref. [128]. Briefly, silicon was first applied, and when completely dried, a C&B MetaBond was applied.

**Single-cell mRNA sequencing**. Cultures produced from 9-day-old embryos, grown for a week DIV, had their medium replaced with dissection-dissociation medium supplemented with 20 U/ml papain. The plate was incubated for 10 min at 37 °C, 5% $CO_2$. Cells were then pipetted thoroughly and moved to a 15 ml canonical tube for gentle centrifugation (5 min at 300 rcf). The supernatant was removed and replaced with 1 ml DMEM with 10% FBS. Cells were then filtered through a cell strainer (40 µm nylon filter, LifeGene). Cells were then centrifuged again, and the medium was replaced as before. Cells were counted using a hemocytometer. Survival assay was performed with trypan blue. Cells were diluted to obtain 1200 cells/µl and 40 µl were processed with 10x scRNAseq with Next-GEM v3 (10x Genomics), analyzed further by cell ranger (10x Genomics) and the Seurat pipeline (Version 4)[62,129–134].

**Library preparation and data generation for single-cell mRNA sequencing**. One RNA single-cell library was prepared according to the 10X manufacture protocol (Chromium Next GEM Single Cell 3' Library & Gel Bead Kit v3.1, PN-1000268) using 20,000 input cells. Single-cell separation was performed using the Chromium Next GEM Chip G Single Cell Kit (PN-1000120). The RNAseq data were generated on Illumina NextSeq2000, P2 100 cycles (R1-28bp, R2-90bp, I1-10bp, I2-10bp) (Illumina, cat no. 20046811).

**Data processing bioinformatic analysis**. Data was analyzed the data using R (Version 4.0.1) and the Seurat R package. Following QC results, we excluded from the data set cells with <200 and >2500 genes (potential cell duplets) and a mitochondrial gene percentage of >30%. In addition, genes detected in less than three cells were filtered out. Following these steps, 9561 cells were left for downstream analysis. Subsequently, PCA was used with all genes as the input and identified significant principal components (PCs) based on the ElbowPlot function. Ten PCs were selected as the input for uniform manifold approximation and projection (UMAP). Cells were clustered by the FindClusters function selecting a resolution of 0.5, yielding 15 different clusters (#0–14). Expression levels were determined based on the $\log_2$ fold change of various differentially expressed genes.

**Statistics and reproducibility**. All data are presented as mean ± SEM. The number of cells are indicated by $n$, and the number of experiments by $N$. Statistical significance (Sigmaplot 11 or Prism 8) was obtained by Student $T$-test (for two-group comparison) or one-way ANOVA for multiple group comparison with post hoc Tukey test. *$p < 0.05$; **$p < 0.01$, and ***$p < 0.001$; n.s., non-significant.

**Reporting summary**. Further information on research design is available in the Nature Portfolio Reporting Summary linked to this article.

## Data availability

All datasets were deposited in a public data repository (https://doi.org/10.5061/dryad.vq83bk3xx; single-cell mRNA seq. data at NCBI GEO, accession GSE227334). All other data and/or materials are available from the corresponding author on reasonable request.

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

## Acknowledgements

We would like to thank Prof. Watanabe D (Department of Biological Sciences, Graduate School of Medicine, Kyoto University) for providing us with the vectors to produce the A3V virus. The research submitted is in partial fulfillment of a doctoral degree for S.Z. Funding— Support was provided by the Rappaport Family Thematic grant (S.B. and Y.G.).

## Author contributions

S.Z.: Data curation, software, formal analysis, visualization, writing—original draft, writing—review and editing; M.A.: Data curation and formal analysis; I.C.: Data curation, formal analysis; R.H.: Data curation, project administration; S.R.: Data curation; Y.G.: Conceptualization, resources, data curation, supervision, funding acquisition, validation, investigation, and writing—review and editing; S.B.: Conceptualization, resources, data curation, software, formal analysis, supervision, funding acquisition, validation, investigation, visualization, methodology, writing—original draft, project administration, and writing—review and editing.

## Competing interests

The authors declare no competing interests.
