## [Peer Review File · Communications Biology]

Reviewers' comments:

Reviewer #1 (Remarks to the Author):

In "A novel AAV1 variant for transduction of Japanese quail neurons in vitro and in vivo" Zoabi et al. have designed a novel AAV variant for more efficient transduction of quail neurons compared to commercially available constructs that are widely used in rodent neuroscience and more recently also in zebra finches (Roberts et al. 2012), pigeons (Rook et al. 2021) and owls (Thiele et al., 2020). To test their self-designed construct, they first established a neuronal cell culture for quail neurons and compared the efficiency of the designed AAV with commercially available constructs, both in vitro and in vivo. Although commercially available viral tools including wildtype AAV1 are already efficiently used in zebra finches (Xiao et al., 2018), pigeons (Rook et al., 2021) and owls (Thiele et al., 2020), to my knowledge they have never been used in quails so far. Thus, in this regard, the study is novel as it tests AAVs in quails for the first time. Moreover, I find the study of great interest and agree with the authors that the avian field in fact will benefit from novel viral tools.

However, at least for other bird species (pigeons, zebra finches and owls) commercially available tools work efficiently enough to transduce cells in general to perform methods such as calcium imaging (Roberts et al., 2017), optogenetics (Roberts et al., 2012; Rook et al., 2021) and genetic lesions (Roberts et al., 2017). Thus, the benefit from novel tools for birds in general would rather come from cell type specificity, improved anterograde or retrograde transport and faster transduction/expression times, Zoabi et al. unfortunately did not assess whether their novel AAV1* results in faster transduction or more efficient anterograde and retrograde transgene expression compared to the wildtype AAV1. In pigeons (Rook et al., 2021), owls (Thiele et al., 2020) and zebra finches (Roberts et al., 2012; Xiao et al., 2018) very long expression times of around 4 - 6 weeks have been reported, which is longer than transduction in rodents. Thus, the avian field in general would largely benefit from faster transduction, as described for AAV/DJ in zebra finches (Düring et al., 2020). Moreover, when it comes to anterograde and retrograde transgene expression wildtype AAV1 is also efficiently used in pigeons (Rook et al., 2021) and zebra finches (Xiao et al., 2018). Thus, it would have been very useful and insightful had Zoabi et al. investigated their novel tool also with respect to these properties, both in vivo and in vitro, to provide a new tool that attracts not only labs working with quails, but the bird community in general.

Moreover, I am not completely convinced that the AAV1* variant designed by Zoabi et al. is superior compared with the wildtype AAV1 (even in quails) and will outline my more specific concerns in the following passages. My main concern is that more expression (as shown for AAV1* compared to AAV1) does not necessarily imply that the construct is more useful for applications such as optogenetics and calcium imaging. More expression can also lead to cell death as for example reported for AAV9 in rodents (Ciesielska, et al., 2013), primates (Samaranch et al., 2014) and pigeons (Rook et al., 2021). To exclude the possibility that the improved transduction goes along with cell death, immunohistochemical stainings against NeuN, GFAP or IBA1 could be performed for the brain slices that were obtained in the vivo injection experiments (just as done in Ciesielska, et al., 2013 and Rook et al. 2021). My concern was raised by Supplementary Figure 8. Here Zoabi et al. describe that cells highlighted with arrow heads are non-neuronal cells. However, as no counterstainings have been performed (such as NeuN (for neurons), GFAP (for astrocytes) or IBA1 (for microglia)), these cells could actually also represent dead cells, especially as their diameter seems to be unnaturally large. In pigeons, expression patterns like this were related to increased GFAP levels and neuronal cell loss (Rook et al., 2021).

In the following I will express my more specific remarks in a chronologically order.

More specific points:

Results

Line 68

The authors do not report that AAVs are also effectively used in zebra finches (Roberts et al., 2012; Xiao et al., 2018) and only report their efficacy for pigeons and owls.

Line 74

The authors say that wildtype AAV1 resulted in extremely variable results even when the same virus batch was used. The findings for this can be seen in Supplementary Figure 1. Only one out of four quails was found to have YFP expression. Were all quails transcidentally perfused? Were there differences in perfusion quality? Native fluorescence signals might be affected by perfusion quality. Counter stainings against YFP either with fluorescence antibodies or DAB stainings (Rook et al., 2021) could increase the signal and might be more accurate for equal comparisons.

Line 182

Different constructs were tested in vitro. How often was every construct tested in vitro?

Line 224

The authors describe AAV1* superiority compared to the wildtype AAV1 in vitro. The authors report percentages of signal increase and p-values. However, SD, the test statistics (t or F values) and the specific test calculated are not reported. Likewise, in the corresponding Figure 6 no SEM or SD are represented. In Line 501 authors say that everything was calculated either with t-test or ANOVA. However, both tests require normally distributed data for which SD can always be provided. Was a Chi square test used? Could the authors elaborate on how the statistics were performed.

Line 227

The authors say that AAV1* retains its tropism towards non-neuronal cells in vitro. This is kind of surprising as AAV1 has shown a natural tropism for neurons in in vivo experiments in birds and mice (Rook et al., 2021, Taymans et al., 2007). The authors also report a tropism of AAV1 and AAV1* for neurons in their in vivo experiments based on visual inspection of cell morphology (as indicated in Line 265). What is the authors idea for the tropism of AAV1 and AAV1* for glia cells in vitro? Are cell culture experiments reliable in this case when the natural tropism between in vivo and in vitro experiments is reversed?

Line 228

The authors state that a lot of cells showed atypical electrical activity and were silent. Could this reflect possible toxic effects of AAV1*?

Line 254

The authors describe AAV1* superiority compared to the wildtype AAV1 in vitro. The authors report percentages of signal increase and p-values. However, SD, the test statistics (t or F values) and the specific test calculated are not reported. Likewise, in the corresponding Supplementary Figure 7 no SEM or SD are represented. In Line 501 authors say that everything was calculated either with t-test or ANOVA. However, both tests require normally distributed data for which SD can always be provided. Was a Chi square test used? Could the authors elaborate on how the statistics were performed.

Line 259

The authors say that they have injected AAV1* expressing either CAG-eYFP or CAG-GCaMP6m into the Wulst of two (4-5 weeks old) and three adult quails (2 month old). Could the authors provide a table providing information about the specific construct that was injected into each specific animal. At the moment I understand that the authors performed $n = 5$ injections into 5 separate animals (I guess only in one hemisphere?) but these 5 animals varied in ages (4-5 weeks or 2 month) and also in the constructs that were injected (CAG-eYFP or CAG-GCaMP6m). However, the results were later pooled for statistical analysis? Did the animals, where AAV1 wildtype

was injected, differ in the same way? It is possible that both age but also the transgene of interest can affect the transgene expression. Thus, it would have been more accurate to keep these factors constant especially when two viral vectors should be compared, and the number of tested subjects is rather small.

Line 261

After seven to eight weeks brains were removed, fixed and sectioned. However, in line 429 authors describe that all animals were transcardially perfused. Was the fixation method kept constant for all animals? And which one was it? Did the authors also use NaCl prior to the fixation with PFA?

Discussion

Line 296 and Line 312

AAV1 also works in zebra finches (Xiao et al.,2018).

Line 331

Its true that so far higher titers have been used in birds compared to this study. However, the authors should be careful with the conclusion that AAV1* can be used with lower titers compared to wild type AAV1 as these comparisons also need to take into account the number of transduced cells. Just because higher titers were previously used does not mean wildtype AAV1 in other species would be ineffective at the same titers that were used for AAV1* in this study. It could simply reduce the number of transduced cells.

Methods

Line 425

The authors assessed the titer of wildtype AAV1 and the self-designed AAV1*. AAV1* had a 1.68 times higher titer than wildtype AAV1. Could the difference in transduction that was observed in the in vivo and in vitro experiments also be the result of the increased titer?

Line 467

The authors injected 0.5 μ l for the in vivo experiments. The general poor infectivity of the in vivo experiments might be the result of the rather small injection volumes. Volumes up to 5 μ l are reported for pigeons and owls (Rook et al., 2021, Thiele et al., 2020) and lead to stable transgene expression without cell death. Maybe the authors can improve their in vivo transduction by increasing their injection volume.

Statistical Analysis

Line 499

The authors say that all data is presented with mean \pm SEM. Although the figures included this information, the information is missing in the result text.

The authors state that they used either t-test or one-way ANOVA for their statistical results. However, for the cell culture experiments the statistics were unclear.

Figures and Figures Legends

Line 783 (Figure 1)

Authors say that the cultured cells remained viable for up to 30 days, but pictures are only shown until day 21 (Figure 1).

Line 851 (Figure 7)

The authors say that 1 μ l of AAV1* or AAV1 was injected. In the methods in Line 467 they write that 0.5 μ l were injected. Was the injection volume kept consistent for all injections? Did it vary between

tested constructs? Did the authors inject 0.5 μ l or 1.0 μ l? How was injection volume controlled in the in vivo injections? Which device was used for injections?

Line 852 (Figure 7)

The authors say that white arrowheads indicate astrocytes. How can the authors be sure that these are astrocytes and not dead cells? The cells look very large. I would propose that NeuN and GFAP stainings should be performed to clarify whether these are indeed astrocytes or rather dead cells, which would indicate cytotoxicity of AAV1*.

Line 862 (Supplementary Figure 1)

The authors describe that four animals have been injected with wildtype AAV1. For the analysis they only used three (see Figure 7). The authors should describe why one animal was excluded from the analysis. The corresponding result section for Figure 7 should include, mean \pm SD, test statistic (t and p values) and the test that was calculated.

Line 866

Coverslips (Typo)

Line 884 (Supplementary Fig. 5)

The figure description does not match the Figure

Line 900 (Supplementary Fig. 7)

Non-neuronal cells could also be dead cells. GFAP and NeuN stainings are needed for the histology of the in vivo injections to make sure that AAV1* does not result in cell death.

Reviewer #2 (Remarks to the Author):

In this article, a clear experimental pipeline was followed in order to improve the suitability of adenoviral particles (AAVs) for avian research. The authors start by describing the faulty features of currently available AAVs, along with other viral particles, for performing experimentation in quail cells and neurons. Later on, the authors set up an elegant in vitro system to validate the use of current and new AAVs and describe the types of cells present in the culture by single cell RNA sequencing (pioneer in quail, to my knowledge). They also identify a potential modification of the receptor enabling AAV infection, and they mutated the sequence of the AAV1 virus to make it more suitable for avian experimentation. Finally, they test this new custom-made AAV1* in vitro and in vivo, in quail and chick models, to describe a significant improvement of the quality of the virus for avian research. The development of this new AAV1* tool is of remarkable relevance to the field of neuroscientists working in avian models, who have been missing such improvements for years.

As detailed, this is a good piece of research that conducts experimentation in a logic fashion. All steps are clearly explained and shown in the figures. The interpretation of their results is adequate, and their conclusion are mostly well supported by their data. There are only some comments and suggestions that I would like the authors to address prior publication.

- Although it is true that AAV1* is proven to be an improved version of all other equivalent tools for avian research, its improvement is not as substantial as one could expect after reading the abstract. The percentage of infected cells/neurons after AAV1* infection is higher than with other virus, but still very low when compared to how those virus perform in mammalian models (all experimentation in Fig. 1 over HEK cells demonstrates that any virus is far better in these cells than AAV1* in quail/chick). I do not know whether it is still possible to improve further AAV1*, not through mutagenesis, but maybe through playing with higher titers, for instance? If not possible, I think the authors should mention the objectively relatively low improvement of the virus infectivity early on in the abstract and end of introduction.

- All AAV infections presented in the article, especially those performed in vivo, are very variable and dependant on the particular conditions of each experiment. This is very obvious when comparing different post-hatching quails infected with AAV1* in equivalent conditions: each animal displayed a very variable number of infected cells. This is why I would like to see how better AAV1* performs along with AAV1, together in the same experimental animal. This could be done by switching the reporter gene of one the virus to a red colour.
- After careful reading of the manuscript, I am not sure whether immunostaining against GFP was performed in any of the conditions. In my own experiments (AAV and plasmid electroporation), GFP immunostaining always boosted the quantity and quality of the cells labelled, in occasions by orders of magnitude. It is possible that without this staining, the authors are underestimating the actual number of infected cells in avian experiments. It is true that this GFP staining is not compatible with some types of research (such as in live cells), but still could be interesting for researchers working with virally infected cells after fixation.
- In figure 3, I recommend the authors to improve a bit the presentation of the data, which seems a bit too raw as derived from Seurat. For instance, the order of the clusters in the heatmap should be changed to present similar clusters together (all fibroblasts one after the other, all neuronal types in a consecutive fashion, etc, as it was done in the dot plot), the information about the identity of the clusters should be presented in the UMAP, and some general features regarding the quality of the sample should be presented (nFeatures, nCount RNA, mitochondrial features, for instance). This last point is very interesting to see, because in my experience the cluster 0 presented here is doubtful and may represent low quality (nearly dying) cells that may better not be included in the analysis. The heatmap of this cluster presents as very unspecific pattern of gene expression, typical of low-quality cells, and by no means it represents the genetic profile of proneuronal radial glial cells.
- It is very confusing to read that quail in vitro cultures comprise over 90% neurons after immunostaining quantification (line 102, figures 1 and 6), but the cellular composition after scRNAseq is so much more diverse, with only about 1000 cells (10% of total) being actual neurons. These numbers and proportions actually impact in the interpretation of the quality of the virus, and would like the authors to specify: where these dissimilarities and numbers come from, and how this new interpretation impacts on the actual quality of infection of the AAV1* virus.
- The colours used in Fig 1b and 6a are not good for discriminating different cells. That magenta colour is not well distinguished, neither on a screen nor in print version. I recommend to change it either red or other more visible colour.
- Taxonomic names of species are consistently wrong throughout the article, both in the text and in figures (such as 5c). The rightly spelled name should be capitalized and italicized according to taxonomic convention.
- In line 141 it is stated that NEUROD6 is a pan-neuronal marker, but it actually is only a pan-glutamatergic neuronal marker.
- I am genuinely surprised about the staging selected for performing the in vitro cultures, E9 in quail. This stage is equivalent to E10 in chick (Ainsworth et al., 2010). At this stage, chick neurogenesis is vastly finished in the forebrain of the chick (Tsai et al., 1981), and therefore it is shocking to read in line 300 that the cultures yielded a very large proportion of neural progenitor cells. In my view, this NPCs are nearly vanishing at this late neurogenic point. Would appreciate the authors to briefly discuss on this staging issue.
- Discussion section is a mere recapitulation of results. I think it should better present a confrontation of this new data with already available data in the literature, as well as to present and discuss the controversial points of this research.
- The brand of the NeuN antibody is not specified in Methods.
- Finally, I consider that many figures can be reduced in size, allowing much information from supplementary figures to be included within main figures.

Reviewer #3 (Remarks to the Author):

The manuscript submitted by Zoabi et al. reports the development a unique protocol for culturing primary neurons and glia from quail embryos. The used transcriptomic analysis to determine the cell composition of their culturing protocol. They then use their neuron culture assay to screen various virus for infectivity and obtained poor results. They next designed novel AAV variants (AAV1-T593K; AAV1*) that show improved transduction efficiency of quail neurons in vitro and in vivo. The authors focused only on quail neurons, which limits the potential audience for their findings. The manuscript needs to be rewritten for clarity and better referenced for accuracy. The manuscript currently will mainly appeal to bird neurobiologists, whereas additional information on the AAV1* infectivity of non-neuronal cells would appeal to additional audiences. This manuscript requires major revisions and rewriting to be considered for publication.

Major revisions

- The paper is difficult to read. Too many long, meandering paragraphs.
- The Supplementary Figures are missing.
- Did AAV1* only infect the AAVR+ cells within the quail cultures as predicted? This question is set up by the author's scRNAseq data and proposed route of infection of AAV1*. However, the author's fail to address this key question.
- What non-neuronal cells does AAV1* infect? The high infectivity rate could appeal to scientists in other disciplines such as developmental biology. These experiments could be carried out by immunofluorescence, scRNAseq, or fluorescence in situ hybridization.

Minor revisions

The authors state that "the use of viruses in quails has been primarily, if not exclusively, employed for transgenesis of primordial germ cells (and solely by lentiviruses)^{6,16,25-29}." This is an absurd statement that ignores decades of publications (i.e., RCAS, MoMLV experiments).

49 The shortage (or complete lack) of viral tools is not limited to the quail model, rather is a recurring theme in the avian field³⁰⁻³³. Resultantly, the avian neuroscience community is largely focused on a handful of species (e.g., finches³⁴, pigeons³⁰ and canaries⁷). This statement is missing experiments in chicken, ducks, and barn owls as a start.

The sections 'Development and characterization of primary cultures from Quail embryos' and 'Single-cell mRNA-sequencing of embryonic quail cultures' could be rewritten to better communicate their findings over their troubles.

The authors detected YFP+ cells in ¼ of the infected quail brains do not appear to be statistically significant. It's confusing why the single positive was only detected at week 7.

Table 1 would be better positioned in Supplementary data.

Something is out of place in line 750-751. Seems like part of Table 2, which appears to be scrambled.

Figure 1 is confusing and lacks negative controls. Figure 1a needs scale bars.

The Discussion section quickly meanders (275-283) into past studies instead of discussing the current study.

Figure 4 shows the lack of A3V infection in the quail cultures but does not provide an example of a positive result for the A3V to confirm the A3V is infectious.

Reviewer #1 (Remarks to the Author):

#1 In “A novel AAV1 variant for transduction of Japanese quail neurons *in vitro* and *in vivo*” Zoabi et al. have designed a novel AAV variant for more efficient transduction of quail neurons compared to commercially available constructs that are widely used in rodent neuroscience and more recently also in zebra finches (Roberts et al. 2012), pigeons (Rook et al. 2021) and owls (Thiele et al., 2020). To test their self-designed construct, they first established a neuronal cell culture for quail neurons and compared the efficiency of the designed AAV with commercially available constructs, both *in vitro* and *in vivo*. Although commercially available viral tools including wildtype AAV1 are already efficiently used in zebra finches (Xiao et al., 2018), pigeons (Rook et al., 2021) and owls (Thiele et al., 2020), to my knowledge they have never been used in quails so far. Thus, in this regard, the study is novel as it tests AAVs in quails for the first time. Moreover, I find the study of great interest and agree with the authors that the avian field in fact will benefit from novel viral tools.

We appreciate the reviewer’s positive comments about the potential benefits of our work to the avian field.

#2 However, at least for other bird species (pigeons, zebra finches and owls) commercially available tools work efficiently enough to transduce cells in general to perform methods such as calcium imaging (Roberts et al., 2017), optogenetics (Roberts et al., 2012; Rook et al., 2021) and genetic lesions (Roberts et al., 2017).

We agree with the reviewer that some AAVs work with select bird species (see page #13), which we refer to in our work (per reviewer suggestion- we have now added more examples- pages # 3 and 13). In fact, previous successes are behind our motivation to examine AAVs in quails (even though all previous viruses employed in quails are lentiviruses). However, all AAV serotypes examined in this work (in particular, variants employed in previous reports, namely AAV1 and AAV2) did not yield efficient transduction of quail brain cells (**Fig. 4**). This was not completely unexpected, as even in the case of AAVs used in published reports to transduce other birds, these do not match the performance as regularly obtained in mammals, rodents in particular. It is reported (and documented by many researchers in the field) that for positive transduction in birds, the AAVs employed require *special* conditions, such as very high titers (10^{13}), prolonged transduction periods (more than 8 weeks), very strong promoters (i.e., CAG), etc. We mention this only to reflect the difficulty in using mammalian-based viruses for infection of bird cells. We therefore believe that our report makes a significant step forward by showing the unique design of a viral vector that exhibits higher transduction efficiencies *in vitro* and *in vivo*. We further emphasize that we employ a new (and much simpler) strategy to develop the new capsid, instead of using very complicated viral-evolution methods or by changing the AAVR (which would require animal-transgenesis). We show that this strategy proved successful to increase AAV1’s transduction efficiency in quail brain cells, *in vitro* and *in vivo* (**Figs. 4, 6, 7 and Suppl. 8**), as well as chicken cells *in vitro* (**Suppl. 7**).

#3 Thus, the benefit from novel tools for birds in general would rather come from cell type specificity, improved anterograde or retrograde transport and faster transduction/expression times, Zoabi et al. unfortunately did not assess whether their novel AAV1* results in faster transduction or more efficient anterograde and retrograde transgene expression compared to the wildtype AAV1. In pigeons (Rook et al., 2021), owls (Thiele et al., 2020) and zebra finches (Roberts et al., 2012; Xiao et al., 2018) very long expression times of around 4 - 6 weeks have been reported, which is longer than transduction in rodents. Thus, the avian field in general would largely benefit from faster transduction, as described for AAV/DJ in zebra finches (Düring et al., 2020). Moreover, when it comes to anterograde and retrograde transgene expression wildtype AAV1 is also efficiently used in pigeons (Rook et al., 2021) and zebra finches (Xiao et al., 2018). Thus, it would have been very useful and insightful had Zoabi et al. investigated their novel tool also with respect to these properties, both in vivo and in vitro, to provide a new tool that attracts not only labs working with quails, but the bird community in general.

Anterograde/retrograde transport is indeed an important feature when using viruses and of potential use to many researchers. However, this added feature was not our main objective, instead our main goal was to initially find/produce a viable virus for quail cells. We focused our attention on the development of a novel variant which required the development of a culturing protocol, characterization of cells via staining, electrophysiology, transcriptomics, etc. We suggest that these demonstrations should be of interest to many avian researchers, from developmentalists, through comparative biologists to neuroscientists alike. Thus, we think that the features we describe and the virus we have produced are an important steppingstone towards further characterization and, perhaps, even subsequent optimization of AAV1*, which should appeal to many researchers not only working on quails.

#4 Moreover, I am not completely convinced that the AAV1* variant designed by Zoabi et al. is superior compared with the wildtype AAV1 (even in quails) and will outline my more specific concerns in the following passages. My main concern is that more expression (as shown for AAV1* compared to AAV1) does not necessarily imply that the construct is more useful for applications such as optogenetics and calcium imaging. More expression can also lead to cell death as for example reported for AAV9 in rodents (Ciesielska, et al., 2013), primates (Samaranch et al., 2014) and pigeons (Rook et al., 2021). To exclude the possibility that the improved transduction goes along with cell death, immunohistochemical stainings against NeuN, GFAP or IBA1 could be performed for the brain slices that were obtained in the vivo injection experiments (just as done in Ciesielska, et al., 2013 and Rook et al. 2021). My concern was raised by Supplementary Figure 8. Here Zoabi et al. describe that cells highlighted with arrow heads are non-neuronal cells. However, as no counterstainings have been performed (such as NeuN (for neurons), GFAP (for astrocytes) or IBA1 (for microglia)), these cells could actually also represent dead cells, especially as their diameter seems to be unnaturally large. In pigeons, expression patterns like this were related to increased GFAP levels and neuronal cell loss (Rook et al., 2021).

We agree with the reviewer that higher expression of proteins may lead to cell damage. However, we do not find that overexpression of YFP (or GCaMP6m) caused any cell damage, whether in our *in vitro* or *in vivo*):

[*in vitro*] 1- Positively-infected cells have a healthy resting membrane potential and, in the case of neurons, exhibit normal electrical activity (**Fig. 2, 6c, d and Suppl. 3**); 2- Positively-infected cells expressing GCaMP6m show dynamic calcium activity as opposed to very high and stationary calcium levels that are indicative of dead/dying cells (**Fig. 6e-g**); 3- Single cell mRNA sequencing data (which we collected from AAV* infected cultures) shows no indication of dead cells (there are no elevated expression levels of mitochondrial genes; indicative of cellular damage and dying cells^{1,2}) (**Suppl. 4**) (reported on page 5). Importantly, our cells exhibited acceptable mitochondrial genes ratio (10-20%)^{1,2}, on par with reports showing sequencing results of brain cells from chicken embryos³, thereby showing the viability of all cell types in our cultures (**Suppl. 4c**).

[*in vivo*] 4- Per reviewer request, we have now stained brain slices with NeuroTrace. These high-power images are now presented in **Suppl. 10** and show healthy looking YFP-positive neurons, as well as demonstrates that the cells of 'larger diameter' (as noted by reviewer) are not dead cells, rather non-neuronal YFP-positive cells have star like and extensive processes, likely astrocytes (as initially mentioned), with healthy looking processes and somata (**Fig. 10, black arrowheads with white outline**). The high-power images clearly show that the cell show no varicosities, blebbing, or structural deformities, as is typically curated for apoptotic/dying cells. Of note, we tried to stain our slices for GFAP, however we could not obtain any positive GFAP-labelling of quail brain slices (nor in sc. mRNA seq. data), despite the positive labeling of rat astrocytes by the same antibody and procedure (**Suppl. 5a, b- right panel**). This results suggests that the common antibody used for labeling mammalian-GFAP (**Sigma Aldrich, Cat.# 636562**) is not suitable for staining mature astrocytes in quail brains. This is not completely unexpected owing to the large phylogenetic distance between these genes across species (**Suppl. 5c**) and this complements our findings whereby GFAP could not be detected in the embryonic cultures by immunostaining (**Suppl. 5b, left and middle panels**) or by single cell mRNA seq (**Table 2**).

In the following I will express my more specifics remarks in a chronologically order. More specific points:

Results

#5 Line 68

The authors do not report that AAVs are also effectively used in zebra finches (Roberts et al., 2012; Xiao et al., 2018) and only report their efficacy for pigeons and owls.

For the sake of brevity, we did not include all references pertaining to Aves and AAVs. Nevertheless, we did mention the seminal work with finches in the introduction of the previously submitted version (reference #34. Roberts T. 2012, Nat Neuroscience- with AAV9). We have now added the work by Xia et al. 2018 to have also worked on finches with AAV1 (now appearing as reference #35, page 2 and in the first section of results,

page #3), as well as mention reports by others (page #3, “and also other viruses in canaries⁴, finches⁵, and more^{6,7}).

#6 Line 74

The authors say that wildtype AAV1 resulted in extremely variable results even when the same virus batch was used. The findings for this can be seen in Supplementary Figure 1. Only one out of four quails was found to have YFP expression. Were all quails transcardially perfused? Were there differences in perfusion quality? Native fluorescence signals might be affected by perfusion quality. Counter stainings against YFP either with fluorescence antibodies or DAB stainings (Rook et al., 2021) could increase the signal and might be more accurate for equal comparisons.

All animals underwent the exact same procedure of transcardial perfusion prior cryosectioning (and even by the same experienced person). We purposefully did not employ antibodies to strengthen signals in our samples with the intent to see whether levels of expression of YFP or GCaMP (see **Figs. 6, 7 and suppl. 8**) would be sufficient for detection. Based on our experience, this provides a very good proxy for whether these levels of expression would be sufficient for *in vivo*-imaging (in which case, antibodies are irrelevant) (**now noted in page 3**). We have also examined three more quails by injection of AAV1 *wt* (and obtained the same results). These are now added to **Fig. 7 and Suppl. 8 (summarized in Table 5)**.

#7 Line 182

Different constructs were tested *in vitro*. How often was every construct tested *in vitro*? Every experiment was repeated at least three times independently (some five times). These are now explicitly noted in the figure legends.

#8 Line 224

The authors describe AAV1* superiority compared to the wildtype AAV1 *in vitro*. The authors report percentages of signal increase and p-values. However, SD, the test statistics (t or F values) and the specific test calculated are not reported. Likewise, in the corresponding Figure 6 no SEM or SD are represented. In Line 501 authors say that everything was calculated either with t-test or ANOVA. However, both tests require normally distributed data for which SD can always be provided. Was a Chi square test used? Could the authors elaborate on how the statistics were performed.

We apologize for this oversight. We have replotted these data to include the mean \pm SEM and statistics used. Further, throughout the text we made sure to properly report the type of test and p-value for each statistical comparison. A Chi square test was not used because the differences between the quails were not categorical (i.e. infected versus not infected), instead T-test is more appropriate for assessing the differences between the means of the two normally-distributed populations. In the revised version we elaborate on the statistical tests employed in the **Methods** section (page #20).

#9 Line 227

The authors say that AAV1* retains its tropism towards non-neuronal cells in vitro. This is kind of surprising as AAV1 has shown a natural tropism for neurons in in vivo experiments in birds and mice (Rook et al., 2021, Taymans et al., 2007). The authors also report a tropism of AAV1 and AAV1* for neurons in their in vivo experiments based on visual inspection of cell morphology (as indicated in Line 265). What is the authors idea for the tropism of AAV1 and AAV1* for glia cells in vitro? Are cell culture experiments reliable in this case when the natural tropism between in vivo and in vitro experiments is reversed?

We removed the sentence “AAV1* retains its tropism towards non-neuronal cells in vitro”, which, we agree with the reviewer, is misleading and a source of confusion. The reviewer is correct to point that AAV1 is mostly neuro-tropic, however we would like to emphasize that not exclusively—it also infects glia, astrocytes in particular^{8–13} (these are now explicitly discussed in page 14). Our results in quails suggest that this is equally true for AAV1*. *In vivo*, we do find that the great majority of infected cells are indeed neurons, but we also obtain several instances of glia (likely astrocytes and see above our reply to comment #4) (**Suppls. 9 and 10**). These results are difficult to compare directly with the embryonic cultures, as most cells in culture are from a very different developmental stage, at which the majority are actually neuronal-precursor cells (giving rise to glia and neurons alike) (**Fig. 3**). Only a small fraction (<20%) are mature neurons (**Fig. 3c, Suppl. 2e, f**). Thus, the results do not negate one another, rather show tropism of AAV1* (but also of AAV1) towards different cell stages.

#10 Line 228

The authors state that a lot of cells showed atypical electrical activity and were silent. Could this reflect possible toxic effects of AAV1*?

We denoted cells that do not **fire** (even though they did show some membrane currents and changes in potential) ‘electrically silent’. This by no means describes dead cells. In fact, the ‘silent cells’ we have patched display hyperpolarized resting membrane potential (RMP), which indicates healthy cells (**Fig. 6 and compare with Fig. 2 and Suppl. 3**). We now realize that the name “silent cells” is confusing and because these are far from being silent- these so-called “silent cells” do show electrical events (in current clamp mode- voltage changes, in voltage clamp mode- membrane currents, see **Fig. 6d**); highly reminiscent of non-neuronal cells (potentially glia or precursor cells, e.g.,^{14–16}), and exhibit robust and dynamic calcium activities (**Fig. 6e-g**). Thus, we have rephrased and replaced the wording ‘silent-cells’ and provide more details about the properties of these cells (**page #9**). Additionally, our single cell mRNA seq. data shows that cells in culture are highly viable (see reply to comment # 4 and **Suppl. 4a, b and Fig. 2**).

#11 Line 254

The authors describe AAV1* superiority compared to the wildtype AAV1 in vitro. The authors report percentages of signal increase and p-values. However, SD, the test statistics (t or F values) and the specific test calculated are not reported. Likewise, in the corresponding Supplementary Figure 7 no SEM or SD are represented. In Line 501 authors say that everything was calculated either with t-test or ANOVA. However, both

tests require normally distributed data for which SD can always be provided. Was a Chi square test used? Could the authors elaborate on how the statistics were performed. As replied in comment #8 (above), we have replotted these data to include the mean \pm SEM and statistics used. Further, throughout the text we made sure to properly report the type of test and p-value for each statistical comparison. A Chi square test was not used because the differences between the quails were not categorical (i.e., infected versus not infected), instead T-test is more appropriate for assessing the differences between the means of the two populations. In the revised version we elaborate on the statistical tests employed in the Methods section and in legends.

#12 Line 259

The authors say that they have injected AAV1* expressing either CAG-eYFP or CAG-GCaMP6m into the Wulst of two (4-5 weeks old) and three adult quails (2 month old). Could the authors provide a table providing information about the specific construct that was injected into each specific animal. At the moment I understand that the authors performed n = 5 injections into 5 separate animals (I guess only in one hemisphere?) but these 5 animals varied in ages (4-5 weeks or 2 month) and also in the constructs that were injected (CAG-eYFP or CAG-GCaMP6m). However, the results were later pooled for statistical analysis? Did the animals, where AAV1 wildtype was injected, differ in the same way? It is possible that both age but also the transgene of interest can affect the transgene expression. Thus, it would have been more accurate to keep these factors constant especially when two viral vectors should be compared, and the number of tested subjects is rather small.

Per reviewer request, we have added six more animals (three injected with AAV1* and three with AAV1 WT to express CAG-eYFP). Thus, together with the previous data we have two groups of five age matched quails (4-5 weeks old at time of injection). All injections were in the same hemisphere, same volume and same depth. Data, statistics, and full description of the experiments (e.g., # animals, injection volumes, age, expression densities, etc.) are fully shown/described in **Fig. 7, Suppl. 8**) and per reviewer request in a table format (**Table 5**). We believe that these changes in the manuscript strengthen the comparison between the two viral vectors.

#13 Line 261

After seven to eight weeks brains were removed, fixed and sectioned. However, in line 429 authors describe that all animals were transcidentally perfused. Was the fixation method kept constant for all animals? And which one was it? Did the authors also use NaCl prior to the fixation with PFA?

We apologize for this mistake. All animals were transcidentally perfused before removing the brain and the same protocol was repeated in all animals. In the revised manuscript, we changed the wording of the sentence to: " After seven to eight weeks animal were perfused and brains were processed for cryosectioning (see Methods)"

Discussion

#14 Line 296 and Line 312

AAV1 also works in zebra finches (Xiao et al., 2018).

We added this reference to the results (page #3, 13, reference #35). It is also worth noting that the AAV receptor in Zebra-finches is more similar to that of rodents, and thereby very different from that of the quail (see **Fig. 5c**). This may partially explain the differences between the reports which we mention in discussion (page #13).

#15 Line 331

Its true that so far higher titers have been used in birds compared to this study. However, the authors should be careful with the conclusion that AAV1* can be used with lower titers compared to wild type AAV1 as these comparisons also need to take into account the number of transduced cells. Just because higher titers were previously used does not mean wildtype AAV1 in other species would be ineffective at the same titers that were used for AAV1* in this study. It could simply reduce the number of transduced cells.

We agree and have revised the text accordingly to avoid concluding about the effective titers in other species.

Methods

#16 Line 425

The authors assessed the titer of wildtype AAV1 and the self-designed AAV1*. AAV1* had a 1.68 times higher titer than wildtype AAV1. Could the difference in transduction that was observed in the in vivo and in vitro experiments also be the result of the increased titer?

To directly address this concern, in the revised manuscript, we have repeated this experiment by using identical concentrations of new viruses (3.3×10^{12}) and obtained identical results, namely AAV1* is much more efficient in transducing cells in-vitro, not to mention tends to yield higher fluorescence in each cell (**Fig. 7a, b**).

#17 Line 467

The authors injected 0.5 μ l for the *in vivo* experiments. The general poor infectivity of the in vivo experiments might be the result of the rather small injection volumes. Volumes up to 5 μ l are reported for pigeons and owls (Rook et al., 2021, Thiele et al., 2020) and lead to stable transgene expression without cell death. Maybe the authors can improve their in vivo transduction by increasing their injection volume.

We injected 0.5 and 1 μ l in vivo (see summary in **Table 5**). However, after repeatedly trying to inject larger volumes (5 μ l as rightly noted), this volume was not well absorbed and persistently leaked outside the tissue. Moreover, we refrained from large volumes to avoid swelling, hematomas and/or cell damage. Thus, while transduction could indeed be improved by larger volumes (or even repeated injections), for the sake of comparison between AAV1 and AAV1*, small volumes are adequate and still holds true.

#18 Statistical Analysis

Line 499

The authors say that all data is presented with mean \pm SEM. Although the figures included this information, the information is missing in the result text. The authors state that they used either t-test or one-way ANOVA for their statistical results. However, for the cell culture experiments the statistics were unclear.

As noted in comment # 8 and #11: we now properly report the type of test and p-value for each statistical comparison in the in-vivo and the in-vitro experiments.

#19 Line 783

Authors say that the cultured cells remained viable for up to 30 days, but pictures are only shown until day 21 (Figure 1).

We have added figures from an additional experiment in which we grew cells to 30 days in culture (**Fig. 1b**).

#20 Line 851 (Figure 7)

The authors say that 1 μ l of AAV1* or AAV1 was injected. In the methods in Line 467 they write that 0.5 μ l were injected. Was the injection volume kept consistent for all injections? Did it vary between tested constructs? Did the authors inject 0.5 μ l or 1.0 μ l? How was injection volume controlled in the in vivo injections? Which device was used for injections?

We apologize for not being clear enough on this matter, In the revised version we added a summarizing table and clarified the above questions (**Table 5**, and in the methods section "**Stereotaxic viral injections in vivo**", page # 19). Briefly, we have injected 0.5 to 1 μ l using a Nanoject III (Drummond) via glass capillaries (filled with mineral oil) at 50 nl/10 seconds rate.

#21 Line 852 (Figure 7)

The authors say that white arrowheads indicate astrocytes. How can the authors be sure that these are astrocytes and not dead cells? The cells look very large. I would propose that NeuN and GFAP stainings should be performed to clarify whether these are indeed astrocytes or rather dead cells, which would indicate cytotoxicity of AAV1*.

As indicated in our reply to comment # 4 and #10 (above), we do not find that infection by AAV1* caused any cell damage, whether in our *in vitro* or *in vivo* (see **Fig. 6**).

As to the large cells in **Figure 7**, in the revised paper we followed the reviewer suggestion and stained slices from quails infected by AAV1*-YFP with NeuroTrace and GFAP. The NeuroTrace-counterstaining shows normal looking somata (with an intact nucleus) and these coincide with YFP fluorescence (**Suppl. 10**). Cells do not appear granulated, blebby, disrupted, clumpy, lack dendritic varicosities, (or any other indication of death). Non-neuronal (NeuroTrace-negative) YFP-positive cells show multiple star-like processes, likely astrocytes (as initially mentioned). These cells also show no varicosities, blebbing, or structural deformities, as is typically curated for apoptotic/dying cells. The anti-GFAP staining did not work (the antibody is viable- positive labeling of rat astrocytes by the same antibody and procedure; **Suppl. 5b**). This results shows that the common antibody used for labeling mammalian-GFAP (**Sigma Aldrich, cat.# 636562**) is not suitable for staining mature astrocytes in quail brains (**Suppl. 5b, right panel**). This is not completely unexpected owing to the large phylogenetic distance between these genes across species (**Suppl. 5c**) and this complements our findings whereby GFAP could not

be detected in the embryonic cultures by immunostaining (**Fig. 4c, middle panel**) or by single cell mRNA seq (**Table 2 and Suppl. 5a**). Dead cells were only obtained following baculovirus infection (**Fig. 4**).

#22 Line 862 (Supplementary Figure 1). The authors describe that four animals have been injected with wildtype AAV1. For the analysis they only used three (see Figure 7). The authors should describe why one animal was excluded from the analysis. The corresponding result section for Figure 7 should include, mean \pm SD, test statistic (t and p values) and the test that was calculated.

We initially did not include older animals in which we found 0 infection in our statistics to avoid overestimation. We then found that younger animals showed better expression of both viruses, which led us to focus on the latter (which also bypasses the uncertainty of whether 0 infection was obtained due to a 'bad' injection). Thus, for the statistical comparison we only included the young animals, as all injected animals showed expression (we did not see a single miss), and all were age matched (**Fig. 7** and see entire dataset in **Suppl. 8**, and **Table 5** with description of all animals). We also injected older animals with AAV1*, and unlike AAV1-injected adults (**Suppl. 1**), and observed expression of YFP (**Table 5**). We note this, but due to a small number of animals (and because the injections were more far apart) we did not show in main figure. Details about the statistics of **Figure 7** have been included.

#23 Line 866
Coverslips (Typo)
Corrected.

#24 Line 884 (Supplementary Fig. 5)
The figure description does not match the Figure
Mistake was corrected.

#25 Line 900 (Supplementary Fig. 7).
Non-neuronal cells could also be dead cells. GFAP and NeuN stainings are needed for the histology of the in vivo injections to make sure that AAV1* does not result in cell death. See our replies to comments #4, 10 and 21. Briefly, NeuroTrace staining shows that YFP-positive cells are indeed stained (**Suppl. 10**) and some YFP-positive cells are non-neuronal (anti-GFAP antibodies yielded no staining of quails cultures or brain slices, see **Suppl. 4**).

Reviewer #2 (Remarks to the Author):

#1 In this article, a clear experimental pipeline was followed in order to improve the suitability of adenoviral particles (AAVs) for avian research. The authors start by describing the faulty features of currently available AAVs, along with other viral particles, for performing experimentation in quail cells and neurons. Later on, the authors set up an

elegant in vitro system to validate the use of current and new AAVs and describe the types of cells present in the culture by single cell RNA sequencing (pioneer in quail, to my knowledge). They also identify a potential modification of the receptor enabling AAV infection, and they mutated the sequence of the AAV1 virus to make it more suitable for avian experimentation. Finally, they test this new custom-made AAV1* in vitro and in vivo, in quail and chick models, to describe a significant improvement of the quality of the virus for avian research. The development of this new AAV1* tool is of remarkable relevance to the field of neuroscientists working in avian models, who have been missing such improvements for years.

We highly appreciate the reviewer's positive comments about our work and the relevance of our work to the avian field.

#2 As detailed, this is a good piece of research that conducts experimentation in a logic fashion. All steps are clearly explained and shown in the figures. The interpretation of their results is adequate, and their conclusion are mostly well supported by their data. There are only some comments and suggestions that I would like the authors to address prior publication.

We would also like to thank the reviewer for appreciating our work, results and interpretations.

#3 Although it is true that AAV1* is proven to be an improved version of all other equivalent tools for avian research, its improvement is not as substantial as one could expect after reading the abstract. The percentage of infected cells/neurons after AAV1* infection is higher than with other virus, but still very low when compared to how those viruses perform in mammalian models (all experimentation in Fig. 1 over HEK cells demonstrates that any virus is far better in these cells than AAV1* in quail/chick). I do not know whether it is still possible to improve further AAV1*, not through mutagenesis, but maybe through playing with higher titers, for instance? If not possible, I think the authors should mention the objectively relatively low improvement of the virus infectivity early on in the abstract and end of introduction.

It is not surprising that all viruses tested on HEK cells yielded much higher efficiency than in quails (Fig. 1), as these are optimized for mammalian cells, not to mention the fact that HEK cells are notorious for yielding very high expression levels of proteins. Nevertheless, our intention was not to compare expression levels, rather to exploit the fact that HEK cells are easy to infect in order to examine the viability of the viruses produced. That said, we agree with the reviewer that the improvement that we show in quails, in-vivo (a mean of about 5 folds higher density of infected cells) is moderate and that the improved infection is not reaching the levels of infections commonly obtained in rodents. We therefore changed the wording in the abstract and introduction to explicitly **mention** that the improvement is modest (e.g., abstract and page #3). We also agree that it would be interesting to continue trying to improve the infection levels by experimenting with concentrations, volumes, waiting times, etc.,. However the main goal of this research was to check our hypothesis that the quail's mutation in the AAVR is responsible in part for the weak performance of AAV1 in quails. We focused our attention on the development of a novel variant which required the development of a culturing protocol, characterization of cells via staining, electrophysiology, transcriptomics, etc. With this approach we

managed to engineer an AAV1* which significantly improves infection levels in quails. This is a first step for further optimization of the virus usage in quails and hopefully may further inspire other genetic modifications in AAVs to be tested in birds.

#4 All AAV infections presented in the article, especially those performed in vivo, are very variable and dependent on the particular conditions of each experiment. This is very obvious when comparing different post-hatching quails infected with AAV1* in equivalent conditions: each animal displayed a very variable number of infected cells. This is why I would like to see how better AAV1* performs along with AAV1, together in the same experimental animal. This could be done by switching the reporter gene of one the virus to a red colour.

In the revised version, we have substantially improved the comparison between the two viruses by adding data from six more quails. Together, our data now contains two groups of 5 age matched animals. Notably, all injections were of the same construct (CAG-eYFP), in the same hemisphere, same volume and depth (see answer to reviewer 1 comment #12) and viral titers. Our results are consistent with our previous observations— we find that there is a significant improvement (~5-fold) in infection when using AAV1* (**Fig. 7b, Suppl. 9**). We hope the new data and analysis address the concern raised above by the reviewer.

The reviewer's suggestion to mix two types of capsids with different reporter genes (red vs. green) and to inject them together is indeed interesting, however presents several hurdles: 1) It has been shown that the expression levels of different reporter genes can vary quite extensively, even if the same capsid and promoter are used¹⁷. 2) Moreover, the extent of expression is difficult to compare between two different colors (i.e., two different fluorescent proteins), as each color requires different detection settings (gain, laser, PMTs, etc.), and therefore it is difficult to ensure that each color is detected to a similar extent. 3) The two viruses can interact in unexpected ways to eventually affect infection without yielding any expression. As noted above, the infection pathway is very complex and while the interaction with the AAVR is essential, it also requires added means for entry. Thus, AAV1 might actually block entry of AAV1* (without gaining access to cell), or vice versa, and this would underestimate the effect of AAV1* without our ability to distinguish between this or other mechanisms involved. Therefore, the expression levels of a mixture of the two viruses may not reflect properly the relative expression levels of each virus alone. We were therefore reluctant to develop this assay (which would require a long process of calibration and validation before conclusive conclusions could be drawn). We can only assume that these are part of the reason why today's most common means employed to assess infectivity of new AAV variants are as described by us (e.g., ¹⁸⁻²¹), and not as those suggested.

#5 After careful reading of the manuscript, I am not sure whether immunostaining against GFP was performed in any of the conditions. In my own experiments (AAV and plasmid electroporation), GFP immunostaining always boosted the quantity and quality of the cells labelled, in occasions by orders of magnitude. It is possible that without this staining, the authors are underestimating the actual number of infected cells in avian experiments. It is true that this GFP staining is not compatible with some types of research (such as in

live cells), but still could be interesting for researchers working with virally infected cells after fixation.

The reviewer is correct. We did not perform immunostaining for YFP or GFP. We do so deliberately because our main goal was to examine whether we could find or produce a virus that would infect and, subsequently, yield sufficient expression levels of proteins and probes for live imaging of neurons *in vivo* (e.g., GCaMP). We agree that for this reason the expression levels are likely an underestimation, but for the sake of comparison this should be adequate. Moreover, our estimations reflect better what we expect to find in live animal experiments. This point is now emphasized on page #10, top paragraph).

#6 In figure 3, I recommend the authors to improve a bit the presentation of the data, which seems a bit too raw as derived from Seurat. For instance, the order of the clusters in the heatmap should be changed to present similar clusters together (all fibroblasts one after the other, all neuronal types in a consecutive fashion, etc, as it was done in the dot plot), the information about the identity of the clusters should be presented in the UMAP, and some general features regarding the quality of the sample should be presented (nFeatures, nCount RNA, mitochondrial features, for instance). This last point is very interesting to see, because in my experience the cluster 0 presented here is doubtful and may represent low quality (nearly dying) cells that may better not be included in the analysis. The heatmap of this cluster presents as very unspecific pattern of gene expression, typical of low-quality cells, and by no means it represents the genetic profile of proneuronal radial glial cells.

We agree with the reviewer's comment per "rawness" of our presentation of the sc-mRNA seq. data and have changed its presentation (see new **Fig. 3**) to present similar cell types in one cluster (i.e., microglia and fibroblasts), however subclusters are still shown (**Suppl. 4c-e**). We also show quality control features (nFeatures, nCount RNA, mitochondrial features; **Suppl. 4a, b**) and added few additional feature plots per select genes of interest (see **Suppl. 4f, Suppl. 5a** and **Suppl. 6**).

Collectively, we do not find any indication of dead cells in cluster 0 (or any other cluster) as there are no elevated expression levels of mitochondrial genes; indicative of cellular damage and dying cells^{1,2} (**Suppl. 4a, b, noted on page #5**). In fact, our *mt* gene counts is on par with recent reports using chicken embryos³. We further show that most cells express SFRP1 (**Fig. 3d**)—a prominent marker for proneuronal radial glia²², strongly supporting our claim of presence of many progenitor cells.

#7 It is very confusing to read that quail *in vitro* cultures comprise over 90% neurons after immunostaining quantification (line 102, figures 1 and 6), but the cellular composition after scRNAseq is so much more diverse, with only about 1000 cells (10% of total) being actual neurons. These numbers and proportions actually impact in the interpretation of the quality of the virus, and would like the authors to specify: where these dissimilarities and numbers come from, and how this new interpretation impacts on the actual quality of infection of the AAV1* virus.

The *in vitro* quantitation of the fraction of neurons in the culture is based on the use of the NeuroTrace Nissl stain—a commonly employed neuronal dye^{23,24}, which suggests that the majority are neurons (**Fig. 1 and Suppl. 1a**). However, our sc-mRNA seq. data suggests

that most of the cells are neuronal precursors (**Fig. 3 and see reply above**). This implies that NeuroTrace may also stain the latter, although a detailed survey of the literature showed that this stain was not used on embryonic proneuronal cells. These are supported by the transcriptomic data showing the very high abundance of neuronal (or neuronal-like) cells (Cluster #0)— a type of neural committed progenitor cells that precedes neurons, i.e., pro-neuronal Radial glial (**Fig. 3c, cluster #0, see summary in Table 2**)²⁵⁻²⁷. Thus, we have subsequently employed a monoclonal anti-MAP2 antibody (Thermo, cat # MA5-12826, **methods**, page #17) which showed much lower labeling (~18%) of cells in culture, on par with the percentage of mature neurons obtained by the sc-mRNA data (~10%) (**Suppl. 2f**). Thus, we now suggest that 10-20% is a better estimation of mature neurons in our cultures. We have now added these descriptions in the text (pages #4-5 and in discussion). We also mention that NeuroTrace may stain “immature” or proneuronal cells (in the discussion).

#8 The colours used in Fig 1b and 6a are not good for discriminating different cells. That magenta colour is not well distinguished, neither on a screen nor in print version. I recommend to change it either red or other more visible colour.

Corrected in main Figure 1.

#9 Taxonomic names of species are consistently wrong throughout the article, both in the text and in figures (such as 5c). The rightly spelled name should be capitalized and italicized according to taxonomic convention.

Corrected.

#10 In line 141 it is stated that NEUROD6 is a pan-neuronal marker, but it actually is only a pan-glutamatergic neuronal marker.

We certainly agree with the reviewer regarding Neurod6 and apologize for the vague description. We have therefore rephrased it accordingly (page #5), and included appropriate references^{28,29}. It is however interesting to note that quail GABAergic neurons also show significant amounts of the transcript (**Fig. 3c**), though indeed less than glutamatergic neurons (a transcriptome signature that is not common in mammals, e.g.,³⁰⁻³²). These show yet again the existing divergences between neurons across species.

#11 I am genuinely surprised about the staging selected for performing the in vitro cultures, E9 in quail. This stage is equivalent to E10 in chick (Ainsworth et al., 2010). At this stage, chick neurogenesis is vastly finished in the forebrain of the chick (Tsai et al., 1981), and therefore it is shocking to read in line 300 that the cultures yielded a very large proportion of neural progenitor cells. In my view, this NPCs are nearly vanishing at this late neurogenic point. Would appreciate the authors to briefly discuss on this staging issue.

The developmental stage selected for producing the *cultures* is E9, equivalent to E10 (or HH36) in chicks³³. Our rationale behind this staging (we actually tested E7 through E9, **Fig. 1**) was based on reports suggesting that neurogenesis should have been completed by these days^{34,35}, at which there should also small amounts of mature glial cells³⁶. We deemed this ideal, as we were mainly interested in examining the suitability of various AAVs to infect neurons. E7 and E8 we less viable than E9, and we have also tried to

produce cultures from post-hatched chicks, however these persistently did not survive in culture (explicitly mentioned in page #4). Thus, these motivated us to remain with E9 cultures.

Our results show very few mature glia cells in culture (e.g., mature astrocytes and oligodendrocytes are completely absent; **Fig. 3c and Suppl. 5a**), supporting previous reports pertaining to these cells types³⁶. However, we find a very large population of neuronal precursor cells, with a small proportion of mature neurons (969/9561 cells; ~10%) (**Fig. 1, Suppl. 2 and Fig. 3**). These suggest that neurogenesis is not fully completed, but this is in line with reports (in chicken) showing that although neurogenesis peaks at ~E7, it continues past this stage through E12 in various brain regions³⁷. Reports also suggest that radial glia (i.e., progenitors of neurons³⁸) can be seen in later stages, as late as E11. In the revised version we discuss this discrepancy between what we find and what is expected from the literature (Discussion (page # 12)). But we do not pursue this direction and therefore do not provide strong conclusions on the matter because comparison between chick and quail developmental stages is beyond the scope of our paper.

To the best of our knowledge, the only study to perform single cell mRNA analysis for chicken embryos do so for very early embryonic stages (HH7 or E1)³⁵. For instance, we do find SOX2 and PAX6—markers of very early developmental stages as reported for the chicken embryo³⁵, but in very few cells (in many clusters) and at very low amounts (**suppl. 6**). These are very different than the appearance provided in chicken embryo³⁵. Nevertheless, they show the validity of our sequencing data and its suitability to classify cells (based on transcriptomic signatures) as reported by others. These are now mentioned on page #7 and discussed in discussion (page #12).

#12 Discussion section is a mere recapitulation of results. I think it should better present a confrontation of this new data with already available data in the literature, as well as to present and discuss the controversial points of this research.

We have extensively revised the discussion, as well as included all issues raised here and by the other reviewers.

#13 The brand of the NeuN antibody is not specified in Methods.

We apologize for this oversight. We have now added the details in the methods section: source- mouse, clone A60, Millipore CAT# MAB377.

#14 Finally, I consider that many figures can be reduced in size, allowing much information from supplementary figures to be included within main figures.

We have revised multiple figures (**Main Figures 1,5,6,7**) to include additional panels and data.

Reviewer #3 (Remarks to the Author):

#1 The manuscript submitted by Zoabi et al. reports the development a unique protocol for culturing primary neurons and glia from quail embryos. They used transcriptomic analysis to determine the cell composition of their culturing protocol. They then use their neuron culture assay to screen various virus for infectivity and obtained poor results. They

next designed novel AAV variants (AAV1-T593K; AAV1*) that show improved transduction efficiency of quail neurons in vitro and in vivo. The authors focused only on quail neurons, which limits the potential audience for their findings. The manuscript needs to be rewritten for clarity and better referenced for accuracy. The manuscript currently will mainly appeal to bird neurobiologists, whereas additional information on the AAV1* infectivity of non-neuronal cells would appeal to additional audiences. This manuscript requires major revisions and rewriting to be considered for publication.

We agree with the reviewer that the quail may not be as popular as other avian species in the field of neuroscience, however we would like to point to the fact that quails are extensively used in other fields (especially in developmental biology³⁹), and therefore we think that our work should be of interest to a large audience, including neuroscientists. In fact, and as noted in the introduction, this land-dwelling bird offers several unique advantages to study behaviors such as spatial memory and navigation and we hope that our tool will make it possible to start studying these birds with state-of-the-art techniques. Thus, although we report a new virus that provides efficient transduction of quail and chicken brain cells, (which may appeal to a small community of researchers), our newly developed strategy to design new viruses based of known sequences of different AAVRs of various birds should likely appeal to a much larger audience.

Major revisions

#2 The paper is difficult to read. Too many long, meandering paragraphs. Rewrite

We have re-written major parts of the manuscript and hope that it is now easier to read, with more references to cover more achievements in field.

#3 The Supplementary Figures are missing.

We find it extremely unfortunate that the reviewer did not receive the supplementary figures which were available to the other reviewers.

#4 Did AAV1* only infect the AAVR+ cells within the quail cultures as predicted? This question is set up by the author's scRNAseq data and proposed route of infection of AAV1*. However, the author's fail to address this key question. AAV1* infected cells expressing the AAVR, however it also infected other cells not expressing the receptor. This could be explained by the fact that the AAV infection route is quite a complex mechanism that involves multiple other receptors and not only aavr. The entry of AAV to cells is highly complex and involves more than just one receptor (i.e., AAVR).

The reviewer is correct regarding the complexity of the infection route of AAV, and although some AAV1 have been shown to require the AAVR⁴⁰, they also requires other membrane proteins that serve as co-receptors^{41,42} (previously noted on page #8, last paragraph). Notably, we find that the AAVR expresses in a variety of cell types (consistent with its "Low cell type specificity" expression as observed in humans and other mammals <https://www.proteinatlas.org/ENSG00000142687-KIAA0319L/single+cell+type>). We do see a slight preference towards microglia (**Fig. 5**). Nevertheless, even in this cluster, we could detect mRNA of this transcript in only ~30% of the cells (**Fig. 5a**), however this is likely the result of the sensitivity of the sequencing method and thus it is hard for us to determine

whether AAVR-negative cells are indeed negative (i.e., really lacking the receptor) or simply exhibit below-threshold detection levels. We could not assess this via immunostaining as the only available antibody (HPA072692) shows expression of the receptor intracellularly- cytoplasmic and Golgi and is unable to detect receptors on membrane(see: <https://www.proteinatlas.org/ENSG00000142687-KIAA0319L/summary/antibody>)! These features make this antibody unsuitable for our needs. These are now discussed in page 13. We note that we did not want to rely on a correlation (or no-correlation), rather aimed to test for causation by specifically pursuing the AAVR-AAV1 interaction for three main reason: 1- there is no other entity that we could target, as the co-receptors for AAVs are not single targets, rather can be many different types of membrane glycoproteins. In the case of AAV1, its N-linked sialic acid glycoproteins⁴¹; 2- the sequence of the AAVR is extant in avian species (**Fig. 5a, c**), 3- there is a highly detailed structure of AAV1, 2 and 5 interacting with the AAVR, which revealed the identity of the interacting pairs (**Fig. 5b** and **Table 3**). In the revised version we discuss this important point in more detail in the Discussion section (Page #13).

#5 What non-neuronal cells does AAV1* infect? The high infectivity rate could appeal to scientists in other disciplines such as developmental biology. These experiments could be carried out by immunofluorescence, scRNAseq, or fluorescence in situ hybridization. We now provide a detailed map of cells that are infected by the virus *in vitro* by sc-mRNA seq data (new **Suppl. 11**), though note that there is no real preference as reported⁴³ in page 14, top paragraph). with a slight preference towards microglia (discussed in page 13). *In vivo*, however, we find that AAV1* is mainly neuro-tropic (**Suppl. 8 and 10**; and see answer to reviewer 1 comment #4).

Minor revisions

#6 The authors state that “the use of viruses in quails has been primarily, if not exclusively, employed for transgenesis of primordial germ cells (and solely by lentiviruses)^{6,16,25–29}.’ This is an absurd statement that ignores decades of publications (i.e., RCAS, MoMLV experiments).

We apologize for how this statement was understood, as we had no intention to ignore previous achievements. In fact, we are aware of reports using MoMLV⁴⁴ for transgenesis of quails, and now mention this study (in introduction). We are also aware of the RCAS system for transgenesis of chickens (e.g.,⁴⁵), although we could not find any explicit mentions for its use in quails in the literature (we find a single mention in a NIH-website: <https://ccr.cancer.gov/hiv-dynamics-and-replication-program/resources/rcas-system>). Regardless, these are all subtypes of retroviruses (as are lenti. viruses) and this was our intention in the text. Thus, to better reflect the literature, we have rephrased it in the text: “the use of viruses in quails has been primarily, if not exclusively, employed for transgenesis (for instance, by retroviruses such as Lentivirus and MoMLV)” and have added the relevant MoMLV reference (# 30 in text) (page 2).

#7 The shortage (or complete lack) of viral tools is not limited to the quail model, rather is a recurring theme in the avian field^{30–33}. Resultantly, the avian neuroscience community is largely focused on a handful of species (e.g., finches³⁴, pigeons³⁰ and canaries⁷).

We have now removed this statement. Our intention was to say that only few animal models are used with genetic tools and probes, as is commonly employed in modern neuroscience.

#8 The sections 'Development and characterization of primary cultures from Quail embryos' and 'Single-cell mRNA-sequencing of embryonic quail cultures' could be rewritten to better communicate their findings over their troubles.

As noted above (see reply to comment #1), we have re-written major parts of the text (including this section) and hope that it is now easier to read. We have also specifically moderated the description of the hurdles we have encountered.

#9 The authors detected YFP+ cells in ¼ of the infected quail brains do not appear to be statistically significant. It's confusing why the single positive was only detected at week 7. Supplementary 1 reflects our multiple attempts using AAV1*wt* to infect brain cells of quails. We have injected multiple animals with AAV1, and examined expression in different animals along the way, specifically focusing on time stamps that are common in the rodent (3 weeks) and avian field (7-8 weeks). In most instances, we obtained no expression, except for one animal (1/4), in which we also observed a small number of cells. These quails were not included in the statistical comparison between infected cell densities because the experimental conditions and waiting times were different. For the statistical comparison we used 10 age matched young quails (see answer to reviewer 1, #12). We initially hypothesized that the virus was not viable, which we later disproved (we test infectivity of all homemade or purchased viruses on cultured cells and primary neurons. It is also not a matter of injection quality, volumes, brain sample preparations, or imaging settings (etc.), as all of these are done by the same experienced researcher, most times on the exact same day, under identical procedures and settings, etc. (now mentioned in text, page #3). The differences that we thereby observe are indeed statistically significant.

#10 Table 1 would be better positioned in Supplementary data.

Having reduced the text quite extensively, we decided to keep Table 1 in main text.

#11 Something is out of place in line 750-751. Seems like part of Table 2, which appears to be scrambled.

We are sorry for the poor quality of the file. We have contacted the editor and have asked to ensure that you obtain the non-scrambled documents and all supplementary figures.

#12 Figure 1 is confusing and lacks negative controls. Figure 1a needs scale bars.

Based on this reviewer's comment #2, we realize that the reviewer did not obtain the supplementary material and could not see that Suppl. 2 shows how cultures die quickly prior the optimization of the culturing method. We also add another panel showing cultures at day 30 (**Fig. 1b**).

#13 The Discussion section quickly meanders (275-283) into past studies instead of discussing the current study.

We have extensively revised the text, including discussion, to include all issues raised here and by the other reviewers and to provide a more in-depth discussion about our observations, development and comparisons with parallel studies.

#14 Figure 4 shows the lack of A3V infection in the quail cultures but does not provide an example of a positive result for the A3V to confirm the A3V is infectious.

A3V does not infect mammalian cells and at the time of production of this virus (DNAs and protocol obtained from its developers⁴⁶, see acknowledgements, page #21), we did not have chicken cultures to employ as a positive control. Moreover, because of lack of infection we have produced three different viral batches (all of very high quality assessed by titer). We therefore did not proceed with this virus. In the revised manuscript, we address this caveat and note that more experiments are needed to draw the conclusion that A3V does not infect quail cells: “Of note, the lack of infectivity of our cultures by A3V is surprising because of its reported infectivity of chicken brain cells⁴⁷. We do not know the reason behind this, however following trials with three different viral batches, we opted to stop pursuing this variant” (page #8).

References—

1. Ilicic, T. *et al.* Classification of low quality cells from single-cell RNA-seq data. *Genome Biol.* **17**, 29 (2016).
2. Luecken, M. D. & Theis, F. J. Current best practices in single-cell RNA-seq analysis: a tutorial. *Mol. Syst. Biol.* **15**, e8746 (2019).
3. Williams, R. M., Lukoseviciute, M., Sauka-Spengler, T. & Bronner, M. E. Single-cell atlas of early chick development reveals gradual segregation of neural crest lineage from the neural plate border during neurulation. *eLife* **11**, e74464.
4. Cohen, Y. *et al.* Hidden neural states underlie canary song syntax. *Nature* **582**, 539–544 (2020).
5. Düring, D. N. *et al.* Fast Retrograde Access to Projection Neuron Circuits Underlying Vocal Learning in Songbirds. *Cell Rep.* **33**, 108364 (2020).
6. Markowitz, J. E. *et al.* Mesoscopic Patterns of Neural Activity Support Songbird Cortical Sequences. *PLOS Biol.* **13**, e1002158 (2015).

7. Liberti, W. A. *et al.* Unstable neurons underlie a stable learned behavior. *Nat. Neurosci.* **19**, 1665–1671 (2016).
8. Nieuwenhuis, B. *et al.* Optimization of adeno-associated viral vector-mediated transduction of the corticospinal tract: comparison of four promoters. *Gene Ther.* **28**, 56–74 (2021).
9. O’Carroll, S. J., Cook, W. H. & Young, D. AAV Targeting of Glial Cell Types in the Central and Peripheral Nervous System and Relevance to Human Gene Therapy. *Front. Mol. Neurosci.* **13**, (2021).
10. Koerber, J. T. *et al.* Molecular Evolution of Adeno-associated Virus for Enhanced Glial Gene Delivery. *Mol. Ther.* **17**, 2088–2095 (2009).
11. Merienne, N., Douce, J. L., Faivre, E., Déglon, N. & Bonvento, G. Efficient gene delivery and selective transduction of astrocytes in the mammalian brain using viral vectors. *Front. Cell. Neurosci.* **7**, 106 (2013).
12. Wang, C., Wang, C.-M., Clark, K. R. & Sferra, T. J. Recombinant AAV serotype 1 transduction efficiency and tropism in the murine brain. *Gene Ther.* **10**, 1528–1534 (2003).
13. Hadaczek, P. *et al.* Widespread AAV1- and AAV2-mediated transgene expression in the nonhuman primate brain: implications for Huntington’s disease. *Mol. Ther. - Methods Clin. Dev.* **3**, 16037 (2016).
14. Woo, D. H. *et al.* Activation of Astrocytic μ -opioid Receptor Elicits Fast Glutamate Release Through TREK-1-Containing K2P Channel in Hippocampal Astrocytes. *Front. Cell. Neurosci.* **12**, (2018).

15. McNeill, J., Rudyk, C., Hildebrand, M. E. & Salmaso, N. Ion Channels and Electrophysiological Properties of Astrocytes: Implications for Emergent Stimulation Technologies. *Front. Cell. Neurosci.* **15**, (2021).
16. Zhong, S. *et al.* Electrophysiological behavior of neonatal astrocytes in hippocampal stratum radiatum. *Mol. Brain* **9**, 34 (2016).
17. Fagoë, N. D., Attwell, C. L., Kouwenhoven, D., Verhaagen, J. & Mason, M. R. J. Overexpression of ATF3 or the combination of ATF3, c-Jun, STAT3 and Smad1 promotes regeneration of the central axon branch of sensory neurons but without synergistic effects. *Hum. Mol. Genet.* **24**, 6788–6800 (2015).
18. Chan, K. Y. *et al.* Engineered AAVs for efficient noninvasive gene delivery to the central and peripheral nervous systems. *Nat. Neurosci.* **20**, 1172–1179 (2017).
19. Flytzanis, N. C. *et al.* Broad gene expression throughout the mouse and marmoset brain after intravenous delivery of engineered AAV capsids. *bioRxiv* 2020.06.16.152975 (2020) doi:10.1101/2020.06.16.152975.
20. Paulk, N. K. *et al.* Bioengineered AAV Capsids with Combined High Human Liver Transduction In Vivo and Unique Humoral Seroreactivity. *Mol. Ther.* **26**, 289–303 (2018).
21. Deverman, B. E. *et al.* Cre-dependent selection yields AAV variants for widespread gene transfer to the adult brain. *Nat. Biotechnol.* **34**, 204–209 (2016).
22. Esteve, P., Crespo, I., Kaimakis, P., Sardonís, A. & Bovolenta, P. Sfrp1 Modulates Cell-signaling Events Underlying Telencephalic Patterning, Growth and Differentiation. *Cereb. Cortex* **29**, 1059–1074 (2019).

23. Susaki, E. A. *et al.* Versatile whole-organ/body staining and imaging based on electrolyte-gel properties of biological tissues. *Nat. Commun.* **11**, 1982 (2020).
24. Mai, H. *et al.* Scalable tissue labeling and clearing of intact human organs. *Nat. Protoc.* **17**, 2188–2215 (2022).
25. Eze, U. C., Bhaduri, A., Haeussler, M., Nowakowski, T. J. & Kriegstein, A. R. Single-cell atlas of early human brain development highlights heterogeneity of human neuroepithelial cells and early radial glia. *Nat. Neurosci.* **24**, 584–594 (2021).
26. Kaczmarczyk, L. *et al.* Slc1a3-2A-CreERT2 mice reveal unique features of Bergmann glia and augment a growing collection of Cre drivers and effectors in the 129S4 genetic background. *Sci. Rep.* **11**, 5412 (2021).
27. Martínez-Cerdeño, V. & Noctor, S. C. Neural Progenitor Cell Terminology. *Front. Neuroanat.* **12**, 104 (2018).
28. Tutukova, S., Tarabykin, V. & Hernandez-Miranda, L. R. The Role of Neurod Genes in Brain Development, Function, and Disease. *Front. Mol. Neurosci.* **14**, 662774 (2021).
29. Wu, S.-X. *et al.* Pyramidal neurons of upper cortical layers generated by NEX-positive progenitor cells in the subventricular zone. *Proc. Natl. Acad. Sci.* **102**, 17172–17177 (2005).
30. Loo, L. *et al.* Single-cell transcriptomic analysis of mouse neocortical development. *Nat. Commun.* **10**, 134 (2019).
31. Dilly, G. A., Kittleman, C. W., Kerr, T. M., Messing, R. O. & Mayfield, R. D. Cell-type specific changes in PKC-delta neurons of the central amygdala during alcohol withdrawal. *Transl. Psychiatry* **12**, 1–10 (2022).

32. Zhong, S. *et al.* A single-cell RNA-seq survey of the developmental landscape of the human prefrontal cortex. *Nature* **555**, 524–528 (2018).
33. Ainsworth, S. J., Stanley, R. L. & Evans, D. J. R. Developmental stages of the Japanese quail. *J. Anat.* **216**, 3–15 (2010).
34. Tsai, H. M., Garber, B. B. & Larramendi, L. M. 3H-thymidine autoradiographic analysis of telencephalic histogenesis in the chick embryo: II. Dynamics of neuronal migration, displacement, and aggregation. *J. Comp. Neurol.* **198**, 293–306 (1981).
35. Williams, R. M., Lukoseviciute, M., Sauka-Spengler, T. & Bronner, M. E. Single-cell atlas of early chick development reveals gradual segregation of neural crest lineage from the neural plate border during neurulation. *eLife* **11**, e74464 (2022).
36. Long-term primary culture of neurons taken from chick embryo brain: A model to study neural cell biology, synaptogenesis and its dynamic properties | Elsevier Enhanced Reader. <https://reader.elsevier.com/reader/sd/pii/S0165027016000601?token=E3796B7C8C78DFD30D165BB7FF2766199F9CA46D8326143420463D349F8CEB6C696DD462313711E35FD838B91FCE61EC&originRegion=eu-west-1&originCreation=20210623072634>
doi:10.1016/j.jneumeth.2016.02.008.
37. Lever, M., Brand-Saberi, B. & Theiss, C. Neurogenesis, gliogenesis and the developing chicken optic tectum: an immunohistochemical and ultrastructural analysis. *Brain Struct. Funct.* **219**, 1009–1024 (2014).
38. Striedter, G. F. & Beydler, S. Distribution of radial glia in the developing telencephalon of chicks. *J. Comp. Neurol.* **387**, 399–420 (1997).

39. Soliman, S. A. & Madkour, F. A. Developmental events and cellular changes occurred during esophageal development of quail embryos. *Sci. Rep.* **11**, 7257 (2021).
40. Pillay, S. *et al.* An essential receptor for adeno-associated virus infection. *Nature* **530**, 108–112 (2016).
41. Lisowski, L., Tay, S. S. & Alexander, I. E. Adeno-associated virus serotypes for gene therapeutics. *Curr. Opin. Pharmacol.* **24**, 59–67 (2015).
42. Nance, M. E. & Duan, D. Perspective on Adeno-Associated Virus Capsid Modification for Duchenne Muscular Dystrophy Gene Therapy. *Hum. Gene Ther.* **26**, 786–800 (2015).
43. Protein atlas for: ENSG00000142687-KIAA0319L.
<https://www.proteinatlas.org/ENSG00000142687-KIAA0319L/single+cell+type>.
44. Mizuarai, S. *et al.* Production of transgenic quails with high frequency of germ-line transmission using VSV-G pseudotyped retroviral vector. *Biochem. Biophys. Res. Commun.* **286**, 456–463 (2001).
45. Sid, H. & Schusser, B. Applications of Gene Editing in Chickens: A New Era Is on the Horizon. *Front. Genet.* **9**, (2018).
46. Matsui, R., Tanabe, Y. & Watanabe, D. Avian Adeno-Associated Virus Vector Efficiently Transduces Neurons in the Embryonic and Post-Embryonic Chicken Brain. *PLoS ONE* **7**, (2012).
47. Matsui, R., Tanabe, Y. & Watanabe, D. Avian Adeno-Associated Virus Vector Efficiently Transduces Neurons in the Embryonic and Post-Embryonic Chicken Brain. *PLoS ONE* **7**, e48730 (2012).

REVIEWERS' COMMENTS:

Reviewer #1 (Remarks to the Author):

Zoabi et al. have significantly revised their manuscript and have addressed all my comments. My concern of apoptotic cells in the in vivo experiments was addressed by providing supplementary figure 10. The higher power resolution images in this figure show that the infected cells in the in vivo experiments are neurons and astrocytes with healthy looking processes and somata. Moreover, Zoabi et al. have explained why no counterstaining has been performed to quantify the transduced cells in vivo. I agree that this way their data provides a better proxy for in vivo calcium imaging where the use of antibodies would not be possible to increase the signals. Additionally, my methodical questions regarding the tissue processing and statistical questions have been addressed and the related information has been updated in the manuscript. Lastly, Zoabi et al. have added more animals to the comparative transduction analysis of AAV1 and AAV1* thereby strengthening the comparison between the two viral vectors in vivo. In the revised version it is now also clearer which constructs were used under which conditions as they have provided this information in table 5. All told, I believe the work is an important contribution to the field of avian neuroscience.

Reviewer #2 (Remarks to the Author):

The authors made a substantial effort to answer all my queries, and I hope they agree with me that the article has been polished during this review. I have no further concerns and I do consider it is a great article for Communications Biology.

Reviewer #3 (Remarks to the Author):

We appreciate the effort that Zoabi et al. put into their revised manuscript that describes the improved infection of quail neurons a novel AAV1. The manuscript is far easier to read and understand. I feel the authors properly addressed our concerns and those of the other reviewers. We are pleased to support the acceptance for publication of the revised manuscript. We feel the novel culturing method and improved infectivity of AAV1* are important additions to avian neuroscience.